# Dynamic Changes in Fecal Microbial Communities of Neonatal Dairy Calves by Aging and Diarrhea

**DOI:** 10.3390/ani11041113

**Published:** 2021-04-13

**Authors:** Eun-Tae Kim, Sang-Jin Lee, Tae-Yong Kim, Hyo-Gun Lee, Rahman M. Atikur, Bon-Hee Gu, Dong-Hyeon Kim, Beom-Young Park, Jun-Kyu Son, Myung-Hoo Kim

**Affiliations:** 1Dairy Science Division, National Institute of Animal Science, Rural Development Administration, Cheonan 31000, Korea; etkim77@korea.kr (E.-T.K.); kimdh3465@korea.kr (D.-H.K.); junkyuson@korea.kr (J.-K.S.); 2Department of Animal Science, College of Natural Resources & Life Science, Pusan National University, Miryang 50463, Korea; wjrm4862@naver.com (S.-J.L.); kty9884738@naver.com (T.-Y.K.); ggabulzima@naver.com (H.-G.L.); md.rahman@bau.edu.bd (R.M.A.); 3Life and Industry Convergence Research Institute, Pusan National University, Miryang 50463, Korea; g.bonhee@gmail.com; 4National Institute of Animal Science, Rural Development Administration, Wanju 55365, Korea; byp5252@korea.kr

**Keywords:** fecal microbiome, dairy calf, diarrhea, PICRUSt

## Abstract

**Simple Summary:**

The microbiota plays a pivotal role in the metabolism and health of animals. The gut microbiome is dynamically changed by various factors including age, diseases and diet. Before rumen development, most of the gut microbes are found in the colon. However, the diversification and functional changes in gut microbes of neonatal calves are not fully understood. Therefore, the aim of this study was to understand the dynamic changes in the fecal microbiome of pre-weaned calves using metagenomic analysis. We observed dynamic changes in the microbial composition during the pre-weaning period of neonatal dairy calves. In addition, we observed a drastic shift in the gut microbiome during diarrheal disease. Thus, the functions and composition of the fecal microbiome were significantly different between diarrheal cows and healthy cows at the same age. Overall, the study findings provide a strong insight into how aging and diarrhea affect the microbial communities of neonatal dairy calves. The results of study help develop strategies to improve early life gut microbiota being significantly relevant to the health status of dairy cows.

**Abstract:**

Microbiota plays a critical role in the overall growth performance and health status of dairy cows, especially during their early life. Several studies have reported that fecal microbiome of neonatal calves is shifted by various factors such as diarrhea, antibiotic treatment, or environmental changes. Despite the importance of gut microbiome, a lack of knowledge regarding the composition and functions of microbiota impedes the development of new strategies for improving growth performance and disease resistance during the neonatal calf period. In this study, we utilized next-generation sequencing to monitor the time-dependent dynamics of the gut microbiota of dairy calves before weaning (1–8 weeks of age) and further investigated the microbiome changes caused by diarrhea. Metagenomic analysis revealed that continuous changes, including increasing gut microbiome diversity, occurred from 1 to 5 weeks of age. However, the composition and diversity of the fecal microbiome did not change after 6 weeks of age. The most prominent changes in the fecal microbiome composition caused by aging at family level were a decreased abundance of *Bacteroidaceae* and *Enterobacteriaceae* and an increased abundance of *Prevotellaceae*. Phylogenetic investigation of communities by reconstruction of unobserved states (PICRUSt) analysis indicated that the abundance of microbial genes associated with various metabolic pathways changed with aging. All calves with diarrhea symptoms showed drastic microbiome changes and about a week later returned to the microbiome of pre-diarrheal stage regardless of age. At phylum level, abundance of Bacteroidetes was decreased (*p = 0.09)* and that of Proteobacteria increased (*p* = 0.07) during diarrhea. PICRUSt analysis indicated that microbial metabolism-related genes, such as starch and sucrose metabolism, sphingolipid metabolism, alanine aspartate, and glutamate metabolism were significantly altered in diarrheal calves. Together, these results highlight the important implications of gut microbiota in gut metabolism and health status of neonatal dairy calves.

## 1. Introduction

Microbes form the predominant community in the gut of animals and play diverse roles in the supply of nutrients [1], regulation of the immune system [2,3], and morphological development of the intestines [4]. Most importantly, gut microbes degrade and transform carbohydrates, lipids or proteins, into a usable form, which are critical for livestock performance, such as weight gain [5] and milk production [6,7]. During the early life of ruminants, the gut microbiota starts to colonize, and they are sequentially established before weaning [8,9,10]. The established gut microbiome has a significant impact on health status and physiology in ruminants [11]. Thus, dairy cows need to acquire proper microbial composition during the early calf period.

Neonatal calf diarrhea frequently occurs worldwide and accounts for > 50% of total mortality in calves [12], resulting in substantial economic and productivity losses in the dairy industry. Calves who have recovered from diarrhea might show subsequent growth stunting, which affects economic performance [13]. A variety of pathogens, including bovine viral diarrhea virus, bovine enterovirus, Cryptosporidium, *Salmonella* spp., and *Escherichia coli* are involved in the occurrence of calf diarrhea [14]. In other hands, non-infectious stresses such as changes in diet and breeding facility induce diarrheal diseases in neonatal calves. Accumulating literature suggests that gut microbiota is significantly associated with the health status of animals [15,16,17,18,19]. Gut microbial community alterations result in the dysregulation of host immune homeostasis and increased susceptibility to disease [20]. Inversely, various diseases also induce a shift in the gut microbiome in animals [21,22,23]. Previous studies have shown that various gastrointestinal diseases are associated with disruption of the gut microbiota composition [24,25]. However, we do not fully understand for biological connection between diarrhea and gut microbiome in calves.

The aim of the present study is to understand changes of diversity and function in the fecal microbiota of pre-weaned calves through monitoring the fecal microbiome by aging and diarrhea. This study provides a strong insight into how aging and diarrhea affect the microbial communities of neonatal dairy calves.

## 2. Materials and Methods

### 2.1. Animals and Diet

A total of five dairy calves were used in the present study. All calves were delivered vaginally. The average body weight (BW) of the calves was 31.4 ± 4.4 kg at birth. The calves were separated from their mothers within 2 h and fed colostrum within 3 h after birth. We obtained and mixed the colostrum from the general mother(s) cow that can get the colostrum of the farm. Colostrum was stored in −80 °C until feeding. All experimental neonatal calves were fed same colostrum. The calves were then moved into indoor individual pens and offered transition milk in the morning and afternoon until three days of age. The colostrum given to each calf had a similar nutritional quality (protein, 15.9%; fat, 8.55%; lactose, 3.03%) and was provided up to 10% of their BW for the first 3 days. On day 3, all calves were fed 2 L of whole milk (protein, 4.06%; fat, 5.33%; lactose, 4.53%) using calf bottles at 8:00 and 16:30. We obtained and mixed the whole milk from general mother(s) cow. Neonatal calves were fed same whole milk. The amount of milk provided 15% of BW by 30 d and 10% BW by 31 d to 44 d, and then gradually decreased to 5% BW by 45 d to 56 d. Calf starter (Onegi-meal, Woosung, DaeJeon, Korea) and mixed grass hay (50% orchard grass and 50% tall fescue on a dry matter basis) were provided on days 7 and 56 onward, respectively. The chemical composition and intake of the calf starter and mixed hay are presented in Appendix A.

### 2.2. Monitoring of Animal Health

General health status was monitored during the experimental period. We observed the fecal conditions of all experimental calves to determine whether the animals showed diarrheal diseases on a daily basis. We determined the severity of diarrhea using fecal scoring systems with the averages of fecal fluidity and consistency. Monitoring was conducted daily (08:00) for 1–2 h following the method described by Larson et al. [26]. Fecal fluidity was scored using a scale of normal (1), soft (2), runny (3), or watery (4). Fecal consistency was scored using a scale of normal (1), foamy (2), mucous-like (3), sticky (4), or constipated (5). The calves received antibiotic treatments (sulfadimethoxine sodium, 55 mg/kg of BW daily Green Cross Veterinary Products Co. Ltd., Yongin, Korea) when the fecal scores exceeded three for two consecutive days or when there were signs of severe disease (e.g., severe cough). Sick animals that received treatment were excluded from the study. The calves that showed mild diarrhea but did not receive any treatments were used to analyze the gut microbiome. In the present study, four calves showed diarrhea symptoms, which were calf #18014, #18030, #1803, and #1809 at 1, 2, 3, and 4 weeks old, respectively.

### 2.3. Feces Collection

Calf fecal samples were collected using a sterile swab kit (fecal swab collection and preservation system, Norgen Biotek, Ontario, Canada) once a week for 8 weeks. Fecal samples were collected from approximately 5 cm in the rectum area and the samples were placed immediately in the enclosed tube containing preservative. First, fecal collection was performed at 2 days of age (0 weeks). We further collected fecal samples every 7 days after the first sampling (1–8 weeks). Samples were stored at −80 °C until DNA purification.

### 2.4. DNA Purification

DNA was extracted using a PowerSoil^®^ DNA Isolation Kit (Cat. No. 12888, MO BIO) according to the manufacturer’s protocol. Bead beating (0.1 mm, glass beads; Bullet Blender Storm 24, Averill Park, NY, USA) at a speed of 4000 rpm for 30 s was used to homogenize the suspension and mechanically disrupt the bacterial cells. The purity and concentration of the purified DNA were measured with a spectrophotometer (Nanodrop1000, Thermo Fisher Scientific Oxoid Ltd., Basingstoke, UK) and the integrity of the DNA was verified by agarose (0.7%) gel electrophoresis. DNA samples were stored at −80 °C until subsequent processing. The extracted DNA were further sequenced, processed and analyzed by Macrogen (Macrogen Inc., Seoul, Korea).

### 2.5. 16S rRNA Gene Sequencing and Sequence Analysis

The V3–V4 regions of the 16S rRNA gene were amplified by polymerase chain reaction using primers that composed ILMN pre-adapter + Sequencing primer + Specific locus primer V3 (5′-TCGTCGGCAGCGTC + AGATGTGTATAAGAGACAG + CCTACGGGNGGCWGCAG-3′; forward) and V4 (5′-GTCTCGTGGGCTCGGA + GATGTGTATAAGAGACAGG + ACTACHVGGGTATCTAATCC-3′; reverse). Equimolar amounts of the barcoded V3–V4 amplicons were pooled and paired-end sequenced on an Illumina MiSeq PE300 platform (Illumina, Inc., San Diego, CA, USA). Raw sequences were trimmed and filtered using FLASH (v. 1.2.11) [27]. To ensure that any subsequent analysis was accurate, sequences shorter than 400 base pairs were discarded. The filtered reads were clustered as operational taxonomic unit (OTU) sequences at 97% similarity using the CD-HIT-OUT [28] and chimeric sequences were identified and removed using rDnaTools (https://github.com/PacificBiosciences/rDnaTool) (accessed on March 2020). The sequence was classified using the Ribosomal Database Project (https://github.com/PacificBiosciences/rDnaTool). Community diversity was estimated using the Chao1, Shannon, and inverse Simpson indices [29]. Briefly, Chao1 is used to estimate diversity from the abundance number of species in a community and the Shannon index is used to quantify specific biodiversity. The unweighted UniFrac distance method [30] was used to perform a principal coordinates analysis and an analysis of molecular variance was conducted to assess the significant differences among samples using the mothur program (v. 1.35.1) [31].

### 2.6. Phylogenetic Investigation of Communities by Reconstruction of Unobserved States (PICRUSt) and Linear Discriminant Analysis Effect Size (LEfSe) Analysis

The functional gene content of the fecal microbiota was predicted using PICRUSt [32] based on taxonomy obtained from the Greengenes (v.13.5) database [33]. The predicted genes were normalized by the 16S rDNA copy number, and their metagenomic contributions were then hierarchically clustered and categorized using the Kyoto Encyclopedia of Genes and Genomes (KEGG) database [34,35,36]. The LEfSe method [37], based on the Kruskal–Wallis sum–rank test, was performed to identify the difference in normal and NID pathways. For this analysis, the critical threshold of the Kruskal–Wallis test was set to 0.05 and the algebraic linear discriminant analysis score was set to ≥2.5.

### 2.7. Statistical Analysis

All statistical and heat map analyses were performed using GraphPad Prism 7 (GraphPad Software Inc., La Jolla, CA, USA). One-way analysis of variance (ANOVA) was used for statistical analysis about OTUs, Chao1, Shannon and Inverse Simpson and *p*-Values were performed for multiple testing according to the Tukey method. The Kruskall-Wallis test was used for statistical analysis about the relative abundances of fecal microbiota and *p*-Values were performed for multiple testing according to the Dunn method.

## 3. Results

### 3.1. Sequencing Information

A total of 4,871,830 sequences with 400–500 nucleotide read lengths were obtained from the 45 samples. Sample information are shown in Table 1. The number of length-filtered sequences in individual samples ranged from 29,583 to 248,723. As a result of the assembly, the read count of 7,698,089 came out from 45 samples, and each sample ranged from 60,529 to 319,434 (Table 1). The sequences were classified into 29 phyla, 73 classes, 136 orders, 265 families, 562 genera, and 669 species.

Bacteroidetes, Proteobacteria and Firmicutes represented 39.93%, 16.40%, and 40.23% of all filtered sequences, respectively. The remainder of the phyla represented less than 1% of all filtered sequences except for Actinobacteria (1.85%) (Figure 1A). The abundant families that represented more than 1% of all the 4,871,830 sequences were *Bacteroidaceae* (25.47%), *Ruminococcaceae* (19.59%), *Enterobacteriaceae* (8.83%), *Lachnospiraceae* (7.60%), *Paraprevotellaceae* (5.70%), *Lactobacillaceae* (3.17%), *Campylobacteraceae* (1.76%), *Veillonellaceae* (1.54%), *Porphyromonadaceae* (1.18%), *Pasteurellaceae* (1.17%), *Clostridiaceae* (1.16%), and *Alcaligenaceae* (1.04%) (Figure 1B). A total of 7725 operational taxonomic units (OTUs) were calculated at a 97% similarity cutoff in combination across all 45 samples, and the number of OTUs in individual samples ranged from 335 to 1374.

### 3.2. Shifts in the Fecal Microbiome Composition of Dairy Calves during the Pre-Weaning Period

To understand the general variations among samples, the microbial composition transition from weeks 0 to 8 was represented by three-dimensional principal coordinates analysis (PCoA) (Figure 2).

From weeks 1 to 8, there were considerable differences and high similarity within the group clustered together. The microbial composition at week 0 was distinct from that of the other weeks. There was clear separation for samples between 0 and 1 week of age. After 1 week, the composition of gut bacteria gradually changed until 6 weeks of age (Figure 2). However, samples were in similar places from 6 weeks to 8 weeks of age, indicating no significant variations during this period.

Next, we determined the changes in diversity of the fecal microbiome through multiple diversity tests. In line with the principal coordinate analysis (PCoA) results, the OTUs of samples at week 0 were much higher than those at weeks 1, 2, and 3 of age (Figure 3A). From week 4, OTUs gradually increased and reached a similar level of OTU in the 0-week-old samples. There was no difference between week 0 and from week 4. Considerable changes were observed in diversity at the early weeks (from weeks 0 to 5) of the pre-weaning period compared to those at the late weeks (from weeks 6 to 8), as indicated by the Chao1 and Shannon indices (*p* < 0.05) (Figure 3B,C). As for the inverse Simpson index, only week 0 value was higher than that in the other weeks (Figure 3D). Collectively, the composition and diversity of the fecal microbiome changed dynamically during the earlier period of pre-weaning (0–5 weeks of age).

Next, to characterize the major changes in the gut microbial populations of the pre-weaned calves, the relative abundances were calculated using OTUs. The three most abundant phyla during the pre-weaning period were Bacteroidetes, Proteobacteria, and Firmicutes (Figure 4A). At week 0, the fecal microbiome had relatively lower levels of Bacteroidetes but higher levels of other bacteria compared with those in other weeks. The composition of Proteobacteria increased after 6 weeks of age. In general, we did not observe any drastic changes in the gut microbial composition at the phylum level. At the family level, the seven most abundant bacteria (more than 1% in average abundance) were *Bacteroidaceae* (25.01%), *Ruminococcaceae* (10.24%), *Prevotellaceae* (8.33%), *Enterobacteriaceae* (6.77%), *Lachnospiraceae* (5.94%), *Marinilabiliaceae* (4.10%), and *Lactobacillaceae* (4.04%) (Figure 4B). The family *Bacteroidaceae* showed the highest abundance at week 1 and gradually decreased until week 8. The composition of *Prevotellaceae* gradually increased with aging. *Ruminococcaceae* existed without significant changes during the pre-weaning period. The relative abundances of *Marinilabiliaceae* were higher during the early weeks (weeks 1–4) of the life compared to those during the late weeks (weeks 5–7). *Enterobacteriaceae* almost disappeared after 5 weeks.

We further examined the species level of gut microbes in the samples. *Bacteroides vulgatus* and *Bacteroides fragilis* were attributed most to the pattern of changes in the *Bacteroidaceae* family during the pre-weaning period (Figure 4C). The listed bacteria in Figure 4C were the main bacterial species to explain the changes in the corresponding families. For example, *Prevotellamassilia timonensis* was attributed to the changes in the *Prevotellaceae* family during the pre-weaning period; it was found nearly at the end of the experimental period, at week 7. The other two species, *Anaerophaga thermohalophila* and *Bacteriodes thetaiotamicron,* were found between weeks 5 and 7 and week 1, respectively. From the *Bacteroidaceae* family, *Bacteriodes uniformis* and *Bacteriodes massiliensis* were not significantly different either from the beginning to the end of week 8. Furthermore, *Faecalibacterium prausnitzii*, one of the most important commensal bacterial species of the *Ruminococcaceae* family was seen between weeks 3 and 4 of the pre-weaning period; however, the relative abundance was very low. In addition, *Lactobacillus johnsonii*, one of the important probiotic strains, did not show a high relative abundance. *Escherechia fergusonii*, a species of the *Enterobacteriaceae* family, appeared clearly at week 4 but they disappear from week 6. These results showed that the composition of fecal bacteria of calves changed in certain periods.

### 3.3. Changes of Functional Gene Family in Fecal Microbiome during the Pre-Weaning Period

Dynamic changes in the gut microbial composition induce changes in functional gene family enrichment in the gut microbes. Thus, we used PICRUSt to analyze the results at weeks 0, 3, and 7. For level 2 KEGG pathways, replication and repair, translation, carbohydrate metabolism, and energy metabolism were the major pathways in the fecal microbiome (Figure 5A). We observed mild changes in the relative abundance of the categorized genes at this level. Many genes categorized under metabolism at the level 3 stage were altered by aging (Figure 5B,C).

We compared the relative abundance of genes associated with specific pathways at weeks 0, 3, and 7. Compared to that at week 0, the fecal microbiome at week 3 had a higher abundance of genes associated with the metabolic pathways, including amino sugar and nucleotide sugar metabolism, galactose metabolism, glycan degradation, pyrimidine metabolism, sphingolipid metabolism glycosaminoglycan degradation, and energy metabolism. However, the relative abundance of genes associated with the metabolic pathways, including xenobiotics by cytochrome P450, tyrosine metabolism, butanoate metabolism, propanoate metabolism, fatty acid metabolism, and tryptophan metabolism, were decreased (*p* < 0.05) in the fecal microbiome at week 3. The abundance of genes associated with the degradation of nutrients such as naphthalene degradation, caprolactam degradation, benzoate degradation, and lysine degradation also decreased (*p* < 0.05) at week 3. We also compared the enriched pathways in the fecal microbiome between weeks 3 and 7. Compared to that at week 3, the fecal microbiome at week 7 had significantly enriched (*p* < 0.05) metabolism and nutrient synthesis-related pathways, including methane metabolism, aminoacyl tRNA biosynthesis, phenylalanine tyrosine and tryptophan biosynthesis, energy metabolism, and terpenoid backbone biosynthesis. However, the relative abundance of genes associated with metabolic pathways, glutathione metabolism, pentose phosphate pathway, inorganic ion transport and metabolism, sphingolipid metabolism, galactose, and amino sugar, and nucleotide sugar metabolism were decreased (*p* < 0.05) in the fecal microbiome at week 7.

### 3.4. Shift in the Fecal Microbiome during Diarrheal Disease

Of the five calves used in the present study, four calves developed diarrhea at weeks 1, 2, 3, and 4, respectively. To determine whether diarrhea induces changes in the fecal microbiome, we analyzed the microbial abundances of the pre-diarrhea, diarrhea, and post-diarrhea weeks of the diarrheal calves (Figure 6).

The relative abundance of Bacteroidetes was lower (*p* = 0.06) in the diarrhea week than in the pre-diarrhea week but was recovered in the post-diarrheal weeks (Figure 6A,B). However, the relative abundance of Proteobacteria was higher (*p* = 0.04) in the diarrhea week than in the pre-diarrhea week and then returned to the base level in the post-diarrheal week. The abundance of Firmicutes was not influenced by diarrhea.

Next, we compared the fecal microbiome at the family level to identify the group of families associated with diarrhea (Figure 6C,D). In Bacteroidetes, the relative abundance of *Bacteroidaceae* decreased (*p* = 0.06) to a greater extent in the diarrhea week compared to that in the pre-diarrhea week (Figure 6C,D). The relative abundance of *Bacteroidaceae* recovered after diarrhea (*p* = 0.33). In Proteobacteria phylum, the relative abundance of *Enterobacteriaceae* higher (*p* = 0.19) in the diarrhea week than in the pre-diarrhea. The relative abundance of this bacterial family significantly (*p* = 0.01) lower in post-diarrhea weeks compared with those in diarrheal week. Collectively, we observed major changes in the fecal microbiome due to diarrhea.

### 3.5. Changes in Abundance of Functional Microbial Genes Family by Diarrhea

We used PICRUSt to analyze the results in the pre-diarrhea, diarrhea, and post-diarrhea weeks (Figure 7A). The Kyoto Encyclopedia of Genes and Genomes (KEGG) pathway has various categories, and there are the top categories, level 1, and Second-highest, level 2. Level 1 includes metabolism, cellular processes, organic systems, and so forth. At level 1, more than 50% of genes belonged to metabolism, more than 20% belonged to environmental information processing, and 6% belonged to genetic information processing. Although we did not observe significant changes at the level 1 stage in the present study, many genes categorized under metabolism and organismal system at the level 2 stage were altered by diarrhea. For example, the abundance of genes involved in metabolism, including other starch and sucrose, histidine, sphingolipid, arginine and proline, alanine asparate and glutamate, was decreased (*p* < 0.05) during the diarrhea week (Figure 7B). In addition, the abundance of genes involved in the pathways associated with the biosynthesis or degradation of nutrients such as glycosphingolipid biosynthesis globo series, glycosphingolipid biosynthesis ganglio series, glycosaminoglycan degradation, other glycan degradation and phenylpropanoid biosynthesis was also lower in the diarrhea week compared to that in the pre- and post-diarrhea weeks. In contrast, the abundance of genes in the pathways of the sulfur relay system, signal transduction mechanisms, replication recombination and repair proteins, phosphotransferase system PT5, flagellar assembly, two-component system, and transcription factors increased in the diarrhea weeks. The abundance of most of the altered genes had recovered by the post-diarrhea week (Figure 7C).

## 4. Discussion

The gut microbiota plays a pivotal role in regulating the host metabolism and immunity throughout mutualistic symbiosis. This symbiotic relationship is established during the early life of animals; however, the factors that affect the colonization of the gut microbiota in the gastrointestinal tract are not fully understood. Different aged animals have distinct microbial populations, and a wide variety of factors determine these gut microbial communities. Various factors influence the colonization of intestinal microbiota such as the microorganisms of the dams, colostrum feeding, solid feed introduction [10], stress condition, microbial interaction [39], delivery mode, intestinal mucin glycosylation [40], gestational age, and birth canal [41]. The pre-weaning stage, when the transmission from pre-ruminant to ruminant occurs, is a critical period for neonatal calves [10]. During this transitional period, neonatal calves experience various changes in metabolism, physiology, and immunity. Many studies have suggested that microbiota colonization in early life significantly affects these changes [42,43]. Prior to birth, a calf’s stomach is considered sterile; however, colonization of microorganisms begins 24 h after birth [44,45]. The bacterial community undergoes a few vital stages for complete development and colonization before weaning. The primary phase starts on day 2, and is mainly composed of *Proteobacteria*, *Bacteroidetes*, and *Pasteurellaceae* in this phase. During the second phase, which lasts from days 3 to 12, a significant change occurs in the ruminal environment. The pioneer communities from the mother via the colostrum or vagina play a crucial role in developing other microbial communities [8]. Our study observed a very distinct microbial profile between weeks 0 and 1 (Figure 2 and Figure 3). One study showed increased diversity and richness with age in fecal microbiome of dairy calves, regardless of weaning [46]. Another study was conducted to investigate the development of gut (jejunum and ileum) microbiota in sika deer that have a similar GI tract structure to cattle, and the findings were in accordance with our study as they found an increased diversity and richness with age, and a more remarkable microbial composition was observed at week 0 than at weeks 6 and 10 of age [47]. Colostrum feeding might have a significant effect on the initial development of the gut microbiota [48] as colostrum is a powerful source of an array of microbiota that contains around 700 species. Colostrum feeding is always a key management practice for calf health, which helps in increasing the number of *Bifidobacterium* and decreasing that of pathogenic *E. coli* [49]. Firmicutes is the dominant phylum in pre-colostrum and infant oral samples, and breastfeeding is the first microbial source [50]. A study on colostrum feeding showed that calves that received enough colostrum within 6 h after birth had better colonization of gut microbiota than calves who did not receive colostrum. In addition, vaginal delivery is one of the most important factors for microbial exposure in calves from dams. *Enterobacteriaceae* and *B. fragilis* are introduced directly to newborn calves through the feces of their mother. We observed significantly abundant *Enterobacteriaceae* and *B. fragilis* during the early phase of the pre-weaning period (weeks 1 to 4) (Figure 4B,C). Although we could not point out single reason for the drastic shift in fecal microbiome during early week of life, we postulate that colostrum, vaginal birth, salivary microorganisms and the microbiota of the dams contribute to making the difference at week 0.

After pioneer colonization, the gastrointestinal tract undergoes successive colonization of gut microbiota depends on a two-way interactive process between the host and microorganisms. Our study showed changes in fecal microbiome over the 8-week period. Although the relative abundances at the phylum level did not show a consistent pattern over the 8 weeks, the diversity of the fecal microbiome showed a gradual transition (Figure 3), indicating that there were distinctive changes at the species level. OTU and Shannon index (diversity) differ among different aged animals [9,39,44,51,52,53]. Beta-diversity decreases and alpha-diversity increases with age [47,54,55]. Additionally, bacterial, fungal and archaeal OTUs significantly differ among calves at week 8, year 1, and year 2 of age, and the Shannon and Chao1 indices indicate changes in the diversity and richness from weeks 2 to 8 [56]. A previous study on dairy calves demonstrated a gradual increase in the Chao1 index over time from weeks 1 to 7 for the pre-weaning period, and the Chao1 index was higher for the calves that showed higher weight gain [55]. The α-diversity in the fecal microbiome of Simmental calves showed a gradual increase from 0 week to 11 weeks old [57]. Consistent with previous studies, our results showed gradual changes in the Chao1 (richness) and Shannon (diversity) indices of fecal microbiota over time, whereas there were significant decreases in α-diversity from weeks 0 to 1 (Figure 3A–D). The PCoA showed significant dissimilarities in microbial composition from weeks 1 to 5, whereas we found high similarity within the group in fecal samples from the later weeks. Therefore, microbiome changes occurred in the gastrointestinal tract of dairy calves during the pre-weaning period.

Bacteroidetes, Firmicutes, or Proteobacteria are the major phyla found in cows from several studies regardless of age [7,9,10,52,58]. However, there are some inconsistent reports for the most dominant gut bacteria at the phylum level. Proteobacteria and Firmicutes were the most prominent phyla, constituting 90% of the total phyla. A different result was reported by Alipour et al., [59] who found Proteobacteria and Firmicutes only after 24 h of birth. In another study, Proteobacteria and Bacteroidetes made up almost 85% of the total phyla, with *Pasteurelleaceae* (belonging to Proteobacteria) as the dominant family [7]. In our study, Firmicutes and Bacteroidetes accounted for 80% among the 29 phyla, were the dominant phyla over the pre-weaning period (Figure 4A). Unlike our study, Dias et al. [39] showed Bacteroidetes increase in the colon and cecum as the calves aged before weaning.

Our study also found that *Bacteroidaceae* was the major group of fecal microbiomes at the family level during the earlier weeks (weeks 1 to 2) for neonatal calves; however, their abundance gradually decreased from weeks 3 (Figure 4B). Like our study, *Bacteroidaceae* has been reported to comprise 37% of the total family groups during the first week of birth; however, it decreases drastically after week 1 [8]. *Enterobacteriaceae* is a dominant family in the OTUs of all fecal samples of the calves [41]. *Enterobacteriaceae* was found at weeks 3–5 of age; however, it disappeared after week 5 and was no longer identified in the samples during the later weeks [10] as was demonstrated in the present study (Figure 4B). The abundance of *Prevotellaceae* gradually increased over time before weaning (week 8 of age). *Prevotella*, which is responsible for protein degradation, is the dominant genus of Bacteroidetes during week 2 of age [8].

*Escherichia fergusonii, B. vulgatis, and B. fragilis* were the dominant bacterial species during the early weeks of age (weeks 0–5); however, they almost disappeared after week 6 of age. These results are explained as the main cause of Bacteroidaceae and Enterobacteriaceae increasing during 1 to 2 weeks of age. *P. timonensis* drastically increased during the late stage of the pre-weaning period, which explains the increased prevalence of *Prevotellaceae* family. A*naerophaga thermohalophila* was also abundant during the late stage of the pre-weaning period. The cause for these changes in abundance in the bacterial family and species is not clearly understood; however, the diet change and the successive colonization of different microbes have been suggested as possible reasons.

Many studies have shown that dietary intervention significantly affects the gut microbiome communities. Diet drives the establishment of specific bacteria during the neonatal period. When newborn calves are fed only milk, the microbiota that is used to utilize milk nutrients, for instance, *Bacteroides*, Proteobacteria, and *Lactobacillus* start to grow. In contrast, when calves are exposed to a solid feed diet, the microbial composition changes drastically with an increase in the abundance of archaeal communities [60]. Abecia et al. [61] conducted a study on goats to determine whether nutritional interventions influenced early life microbial colonization and found that the archaeal population was largely modified by nutritional intervention.

As a high proportion of dairy un-weaned calves die due to neonatal diarrhea, it is a global problem that leads to significant economic losses in the dairy industry. Diarrhea is caused by multiple reasons rather than a single factor. We do not fully understand the physiological changes in the gastrointestinal tract of neonatal calves suffering from diarrheal diseases. In general, an enteric infection is considered one of the main reasons for early life mortality and diarrhea. Beyond enteric infection, growing evidence suggests that gut microbiota plays an essential role in the occurrence of diarrhea. High diversity and stability of gut microbial communities are indicative of healthy calves [62]. Several studies have demonstrated that diarrhea could be predicted during early life by observing the shift in the microbial dynamics, which provides us with the opportunity to improve calf health [63,64]. Therefore, the unstable and low diversity can be a potential biomarker for predicting diarrhea. More specifically, many studies have suggested that alterations in the gut microbiome, especially the reduction in butyrate-producing bacteria, cause diarrhea in calves during the first week [55,58]. However, dysbiosis-mediated intestinal disease during the pre-weaning period of the dairy calf has not yet been fully investigated. Our study analyzed fecal microbiome data to understand dynamic changes in the composition and function of fecal microbiome during diarrhea (pre-diarrhea, diarrhea, and post-diarrhea). Among five experimental calves, four calves showed mild diarrhea temporarily and recovered at certain week in this study. Interestingly, we detected significant differences in the fecal microbiota composition and its metabolic pathway-related functional genes from diarrhea to recovery. As all calves fed colostrum, calf diet and showed healthy conditions in the most of experimental periods, we assumed that calves had normal gut microbiota community. Although, they had normal gut microbiota, however, they have suffered mild diarrhea due to various extrinsic factors such as diet change during pre-weaning period.

Although it is difficult to define the causative factors such as microbes for diarrheal disease, general trends can be inferred from our study. There was a decrease in Bacteroidetes and *Bacteroidaceae* between the pre-diarrhea and diarrhea weeks in the neonatal calves at the phylum and family levels (Figure 6).

A decrease in Bacteroidetes is related to the presence of diarrhea, which was consistent with a previous study on healthy and diarrheal calves between two farms [26]. Other studies of different animals have shown that diarrhea is involved in a decrease in the phylum Bacteroidetes [55,65,66,67].

The genus *Bacteroides* is related to various host functions such as maturation of the host immune system [68], glycosylation of the gut epithelium [69], and protective effects against inflammation [70,71]. *Bacteroides* species have also been found to be involved in mediating the stability of gut colonization and the symbiosis mechanism via physical interactions with the host [72]. Thus, having normal range of *Bacteroides* may be reliable indicator for a healthy gut.

Our study revealed an increase in the phylum Proteobacteria in the family *Enterobacteriaceae* when calves suffered from diarrhea. *E. coli,* which belong to Proteobacteria, is one of the most significant bacterial agents that cause neonatal diarrhea worldwide during the first 4 days of calf life [73]. With the help of fimbrial antigens, Enterotoxigenic Escherichia coli (ETEC) is increased in the intestine and multiplied instead of being passed through the feces, which alters the gastrointestinal environment and causes dysbiosis [15]. Together with our results, the decrease in the relative abundance of Bacteroidetes results in increases in the Firmicutes:Bacteroidetes ratio, which has been reported in animals with diarrhea [66,67,74]; therefore, dysbiosis of intestinal microbiota accounts for the occurrence of diarrhea. The mechanism by which Proteobacteria accelerates the incidence of diarrhea is not clear. However, several studies have shown that when the abundance of Bacteroidetes is low, animals start producing δ-aminolevulinic acid with the help of the tricarboxylic acid (TCA) cycle using glycine and succinyl-CoA. Metabolites such as δ-aminolevulinic acid act as markers to increase the abundance of Proteobacteria, and the aftermath is dysbiosis [58]. These results demonstrate that Bacteroidetes (*Bacteroidaceae*) and Proteobacteria (*Enterobacteriaceae*) are the most responsible phyla (family) behind this anomaly. A previous study reported that *Cryptosporidum parvum* viral infection increased *Fusobacterium* (14.1%) in the feces of calves [75], another study reported that *Mycobacterium avium* subsp. *paratuberculosis* (MAP) infection increased Actinobacteria (34.6%) in the feces of calves [76] but, we could not observe these changes in this study. This suggests observed altered fecal microbiome was not due to *C. parvum* or MAP viral infection.

We observed differences in the functional gene levels between healthy and diarrheal calves. PICRUSt exploits a reconstructed algorithm to forecast which gene families are present and then accumulates all the gene families to calculate the compound metagenome and determine the abundance of gene families [33]. The genes responsible for the metabolism of porphyrin and chlorophyll are decreased (*p* < 0.05) in the diarrheal calves [58]. At week 2 of age, ATP-binding transporters are very abundant; however, TonB-dependent transporters are available at week 6 of age [42]. In our study, more genes were involved in metabolism and genes were decreased, which is consistent with the results of a previous study on cats and humans [58] that indicated the disturbance of several nutrient metabolisms leads to dysbiosis-related diseases. In hemorrhagic diarrhea, genes related to ribosome translation, amino acid metabolism, and carbohydrate metabolism are significantly decreased [77]. Hence, many studies including our study suggest that the improper metabolism of amino acids is the most responsible factor associated with diarrhea.

Although, we obtained interesting observations in the fecal microbiome from this study, our study had several limitations like other industrial microbiome studies. First, the fecal microbiome does not really represent the whole gut microbiome. Several studies reported that there are differential microbial communities between feces and gut tissue. Although the fecal microbiome can be a potential biomarker to determine the physiological status of animals, fecal microbiome analysis provides limited information for gut microbial community. Therefore, a comparative study between the fecal and gut microbiomes for developing an effective biomarker is necessary for future study. Second, we could not point out a major single reason for the changes to the fecal microbiome during the pre-weaning period. As a variety of factors affect the fecal microbiome, it is hard to find a major cause of microbiome changes. But we believe that a calf feeding program, such as colostrum and a solid diet, could be major driver for a fecal microbiome change. A dietary intervention study is needed, to be performed in future studies. Third, it is unclear whether the incidence of diarrhea causes an imbalance in gut microbial communities or dysbiosis of the gut microbiome initiates diarrheal disease. It seems that the gut ecosystem is disturbed by an imbalance in the gut microorganisms or any enteric pathogens, it will hamper the entire metabolism cycle during diarrheal disease development. Therefore, an integrated and elaborated approaches are required to elucidate the key mechanism of host-microbial interactions during the pathogenesis of diarrhea. For example, performing shotgun sequencing may help to find a causative microbe such as a virus for diarrhea. Also, experimentally-induced fecal microbiota changes by pathogen infection or fecal microbiota transfer can be a useful approach for future study. To minimize the rate of calf mortality, we must follow the strategies for the intervention of gut microbiota via the manipulation of diet and colostrum feeding management for strong mucosal immunity. Collectively, the fecal microbiome undergoes dynamic changes in neonatal calves during the pre-weaning period. During this period, diet (colostrum and solid feed) and disease (diarrhea) can be major factors for fecal microbiome shift.

## 5. Conclusions

This study was designed to understand changes of diversity and function in the fecal microbiome of pre-weaned calves by aging. The fecal microbiome showed higher diversity at day 2 post birth (0 week) and it drastically changed from week 1 to week 8. Neonatal calves undergo changes in the fecal microbiome and a major reason could be diet changes during pre-weaning period. Diarrheal animals also displayed drastic changes, which are characterized by the high abundance of Proteobacteria and the low abundance of Bacteroidetes with various altered metabolic pathway gene abundances. Thus, this study provides a strong insight into how aging and diarrhea affect the gut bacterial communities of neonatal dairy calves.

Our study creates the necessity for conducting further research to address several questions, including “Does fecal microbiota manipulation improve the condition of diarrhea in neonatal calves?” and “What is the impact of the fecal microbiome shift during diarrhea on host metabolism?” Further studies are required to investigate the dynamic interaction between diarrheal disease and the gut microbiome. This information can be utilized for the development of microbial strategies to improve animal health and performance.

## Figures and Tables

**Figure 1 animals-11-01113-f001:**
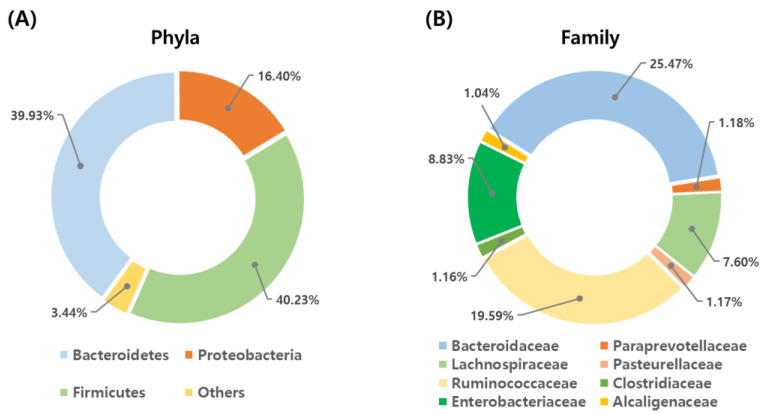
Taxonomic composition of bacterial phyla and family using 16S rRNA gene sequencing. The circular graphs show the percent of sequences assigned to each of the bacterial phlya (**A**) and family (**B**) from calf fecal samples.

**Figure 2 animals-11-01113-f002:**
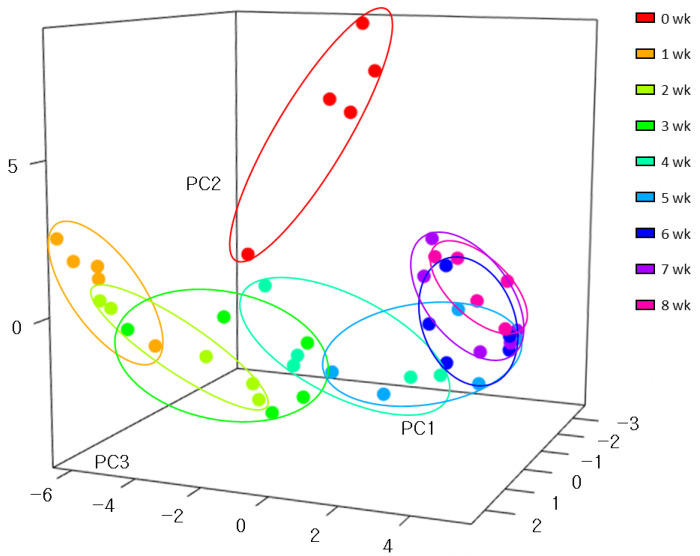
Dynamic changes of fecal microbiome in neonatal calves during pre-weaning stage. Three-dimensional principal coordinate analysis (PCoA) of the fecal microbiome of five dairy calves during the pre-weaning period. The three axes (PC1, 57.95%; PC2, 16.9%; and PC3, 10.01%) of the PCoA explain 84.86% of the variance.

**Figure 3 animals-11-01113-f003:**
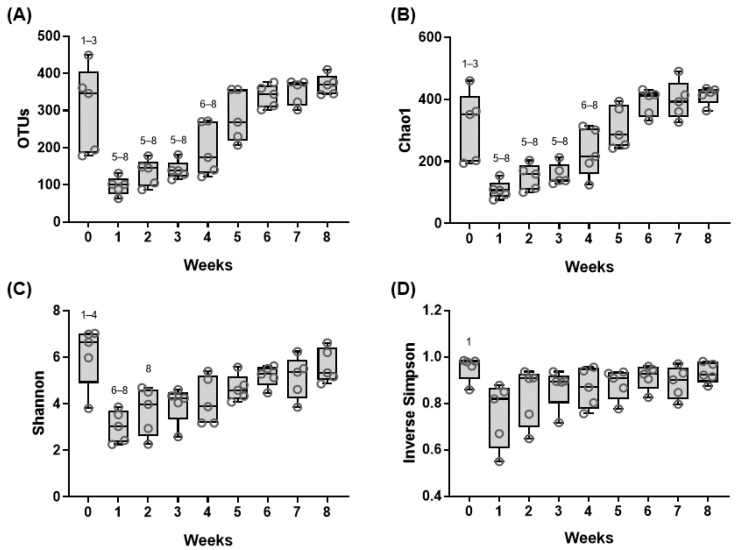
Changes of diversity of fecal microbiome in neonatal calves during pre-weaning stage. Diversity of fecal microbiota in dairy calves during the pre-weaning period. (**A**) Number of observed operational taxonomic units (OTUs), (**B**) species richness (Chao1), (**C**) Shannon, and (**D**) inverse Simpson indices are presented as box-and-whisker plots. Whiskers indicate the mean to max. The numbers above the boxes denote significant differences (*p* < 0.05) to the time points (one-way ANOVA followed by Tukey’s multiple comparisons test).

**Figure 4 animals-11-01113-f004:**
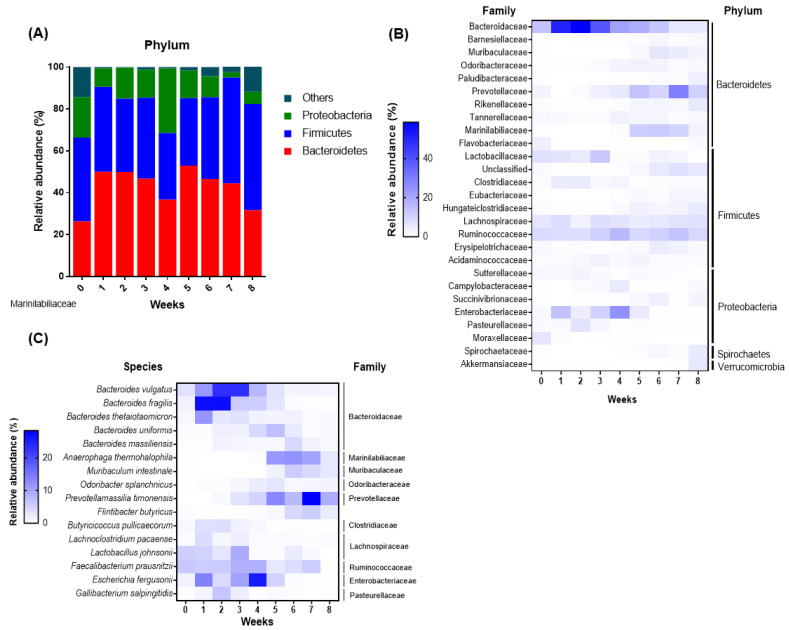
Changes in relative abundance of the fecal microbiota from the dairy calves during the pre-weaning period. (**A**) Relative abundances of the three most abundant phyla (Proteobacteria, Firmicutes, and Bacteroidetes) in the fecal microbiota are presented as stacked bars. (**B**) Relative abundances of families and (**C**) species are presented as heat maps. More than 1% in average abundance of families and species for 9 weeks are shown. Only known species are presented.

**Figure 5 animals-11-01113-f005:**
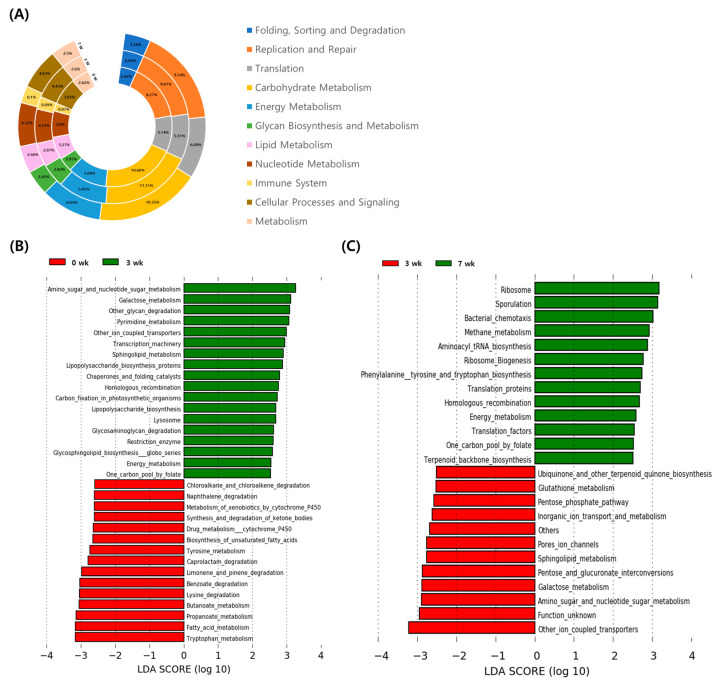
Predicted functional composition of metagenomes based on 16S rRNA gene sequencing data for weeks 0, 3, and 7. (**A**) Graph shows relative abundances of Class 2 Kyoto Encyclopedia of Genes and Genomes (KEGG) pathways at weeks 0, 3, and 7. Linear discriminant analysis (LDA) effect size (LEfSe) performed on Class 3 KEGG pathway data for weeks (**B**) 0 and 3, (**C**) 3 and 7. Grouped data were first analyzed using the Kruskal-Wallis test with a significance set to 0.05 to determine if the data was differentially distributed between groups, and those taxa that were differentially distributed were used for LDA model analysis to rank the relative abundance difference between groups. The LDA for significance was set to ±2.5, and the log(10) transformed score is shown to demonstrate the effect size. Data were analyzed and prepared through Hutlab Galaxy provided through the Huttenhower lab [38].

**Figure 6 animals-11-01113-f006:**
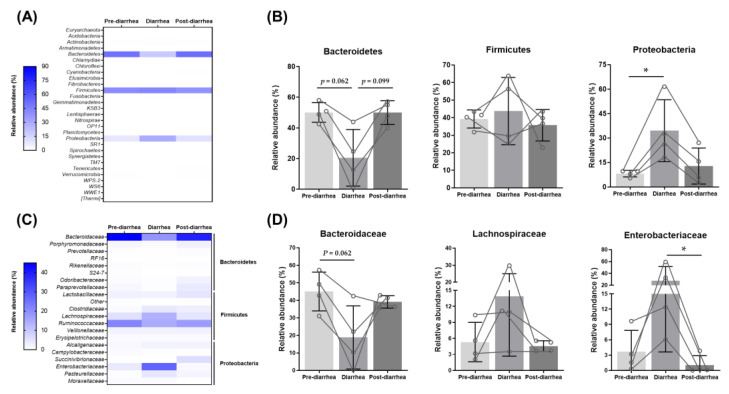
Comparison of the changes in the relative abundances of fecal microbiota in pre-diarrhea, diarrhea, and post-diarrhea weeks. (**A**) Relative abundances of fecal microbiome of diarrheal calves in pre-diarrhea, diarrhea, and post-diarrhea weeks at the phyla level are presented as a heat map. (**B**) Relative abundance of Bacteroidetes, Firmicutes, and Proteobacteria. (**C**) Relative abundances of the fecal microbiome of diarrheal calves in pre-diarrhea, diarrhea, and post-diarrhea weeks at the family level are presented as heat maps. (**D**) *Bacteroidaceae*, *Lachnospiraceae*, and *Enterobacteriaceae* at the family level are shown as bar graphs in the pre- diarrhea, diarrhea, and post-diarrhea weeks. Asterisks denote significant differences between samples (Kruskall-Wallis test followed by Dunn’s multiple comparisons test, * *p* < 0.05).

**Figure 7 animals-11-01113-f007:**
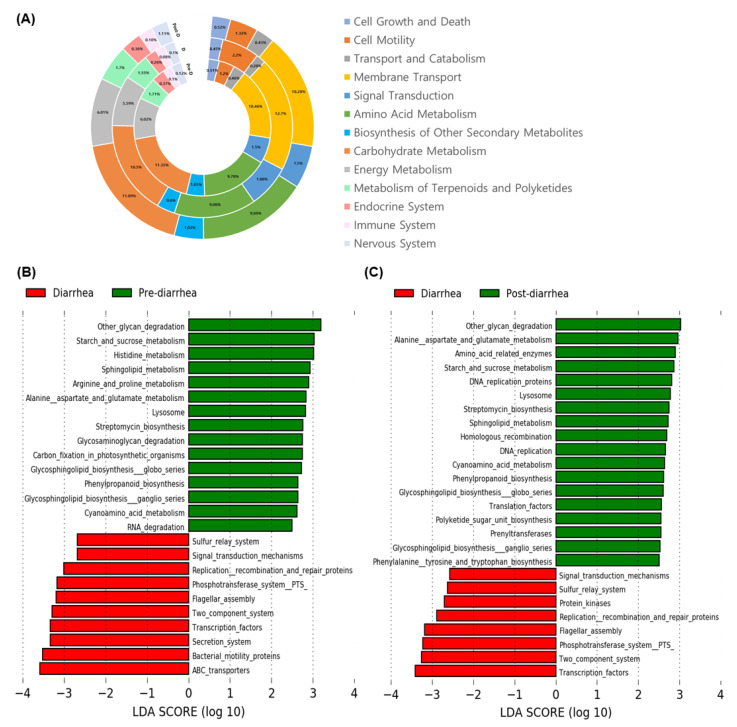
Predicted functional composition of metagenomes based on 16S rRNA gene sequencing data for pre-diarrhea, diarrhea, and post-diarrhea weeks. (**A**) Graph shows relative abundances of Class 2 Kyoto Encyclopedia of Genes and Genomes (KEGG) pathways at weeks of pre-diarrhea, diarrhea and post-diarrhea. Linear discriminant analysis (LDA) effect size (LEfSe) performed on Class 3 KEGG pathway data for weeks (**B**) pre-diarrhea and diarrhea, (**C**) diarrhea and post-diarrhea. Grouped data were first analyzed using the Kruskal-Wallis test with a significance set to 0.05 to determine if the data was differentially distributed between groups, and those taxa that were differentially distributed were used for LDA model analysis to rank the relative abundance difference between groups. The LDA for significance was set to ±2.5, and the log(10) transformed score is shown to demonstrate the effect size. Data were analyzed and prepared through Hutlab Galaxy provided through the Huttenhower lab [38].

**Table 1 animals-11-01113-t001:** Result of Assembly (FLASH) of weekly calf feces.

Sample Name ^1^	Total Bases ^2^	Read Count ^3^	Q20(%) ^4^	Sample Name ^1^	Total Bases ^2^	Read Count ^3^	Q20(%) ^4^
18014.0 wk **^5^**	68,907,327	151,588	97.31	1802.5 wk **^5^**	120,851,766	266,565	98.53
18014.1 wk	73,276,256	161,129	97.88	1802.6 wk	121,996,008	271,757	98.59
18014.2 wk	75,203,142	164,647	97.38	1802.7 wk	127,730,965	283,648	98.43
18014.3 wk	50,686,786	111,548	97.43	1802.8 wk	110,898,306	246,337	98.49
18014.4 wk	65,018,120	143,310	97.36	1803.0 wk	27,571,311	60,529	97.35
18014.5 wk	58,783,722	129,694	97.51	1803.1 wk	58,760,407	128,844	97.45
18014.6 wk	55,635,573	123,136	97.55	1803.2 wk	75,370,647	166,826	97.62
18014.7 wk	46,376,877	102,750	97.67	1803.3 wk	62,276,136	137,975	97.6
18014.8 wk	51,792,102	115,213	97.5	1803.4 wk	77,599,897	172,809	97.55
18030.0 wk	86,128,455	191,030	98.48	1803.5 wk	52,503,714	116,060	97.49
18030.1 wk	73,886,165	163,363	98.35	1803.6 wk	56,767,208	125,325	97.48
18030.2 wk	60,748,672	134,589	98.53	1803.7 wk	56,836,785	126,461	97.6
18030.3 wk	76,840,598	169,738	98.32	1803.8 wk	62,833,274	139,711	97.65
18030.4 wk	81,039,398	178,623	98.36	1809.0 wk	66,532,385	146,064	98.92
18030.5 wk	64,971,503	143,031	97.74	1809.1 wk	71,049,665	154,645	98.66
18030.6 wk	62,737,787	139,009	98.22	1809.2 wk	79,287,856	173,699	98.69
18030.7 wk	88,812,047	197,187	98.38	1809.3 wk	69,925,292	152,643	98.87
18030.8 wk	66,357,601	147,718	98.39	1809.4 wk	71,951,762	158,274	98.68
1802.0 wk	101,856,946	223,217	98.29	1809.5 wk	73,684,048	161,208	98.81
1802.1 wk	126,700,114	280,737	98.66	1809.6 wk	70,792,739	155,381	98.5
1802.2 wk	126,537,869	276,213	98.59	1809.7 wk	71,128,878	156,248	98.51
1802.3 wk	120,722,588	265,181	98.61	1809.8 wk	73,935,709	164,995	98.46
1802.4 wk	144,618,157	319,434	98.81	-	-	-	-

^1^ Sample name, animal object number; ^2^ Total bases, the total number of bases in reads identified; ^3^ Read Count, the total number of sequences reads; ^4^ Q20(%), the percentage of bases in which the phred score is above 20; ^5^ wk, weeks of age.

## Data Availability

All data presented in this study are available on request from the corresponding authors.

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
