# Peer review of "Dynamic Changes in Fecal Microbial Communities of Neonatal Dairy Calves by Aging and Diarrhea"

_animals, 2021, doi:10.3390/ani11041113_

Round 1

Reviewer 1 Report

This is quite a nice and interesting paper investigating the microbiome development of animals over a period of time from birth. I think calling it investigation of diarrhoea is a bit of a push, as it seems more of an after thought. There is a lot of data missing from the methodology which makes it difficult to review in its current form, so I can only recommend modifications and resubmit.

Line 57- remove fibers and it doesn’t add anything

Line 61- I would argue that it is a very dynamic system, which changes constantly. So do you mean it is difficult to completely change? Or destroy? As in the main, it will alter in response to diet, stress, hormones, pregnancy etc

Line 66- maybe say growth stunting rather than disorders- but up to you

Line 66- maybe including bvdv etc? Others include Cryptosporidium

Line 69- link the two sentences

Line 81-83- is this also diet associated? And stress?

Line 83- diarrhoeic calves?

Line 78-89- this is almost a summary and isn’t needed in the introduction. It needs an aims in here instead

Line 98- was this colostrum from the same mother? Or commercial? If the latter please state manufacturer

Line 101- reword as doesn’t make sense

Line 102- was this milk pasteurised or from the cows mother?

Line 103- reword as doesn’t make sense

Line 106- the grass on day 7 is likely to impact the microbiota, and could induce diarrhoea

Line 117-118- please reword as sounds repetitive

Line 120- scoring systems

Line 124-127- this will also have affected the gut microbiota. You say that these were excluded, how many did that include? Based on the data below, 4 have diarrhoea, so they were removed, leaving 1 animal left?

Line 133- where in the sample you took this from is important as that will also affect microbiota

Line 136- sampling every 7 days may miss the diarrhoea completely? So you see an alteration and miss it due to sampling technique?

Line 146-147- how was this further sequenced and analysed? More details needed

Why was 16S rRNA metagenomics chosen over shotgun? Diarrhoea may be of parasitic, protozoan or viral aetiology?

Line 184- this is the first mention of the 45 samples, maybe want to say where these came from.

Line 183-185- this may be useful in table with more info on the samples, numbers of sequences etc

Section 3.1 some figures and tables would make this much clearer to read

Figure 1 looks great

Line 214- is this OTU?

I think that most of the bacterial names need to be in italics

Figure 3, this is a bit small to take much detail from, but that is probably a formatting issue

Line 256- reword

Line 263- please reword this as it sounds like the bacterium causes the disease rather than the family

Line 267- I struggle with this as you only did 16S rRNA gene metagenomics, so how can you comment on functional genes?

Line 301- would be nice to have more details on the gastroenteritis and when it occurs and in which animals, and if treated.

Author Response

Response to Reviewer-1

GENRAL RESPONSE: Thanks for valuable comment. We have improved our manuscript as followed by your kind suggestion. In revision, we focused on method part: 1) Animal Feeding information 2) Fecal sample Collection. Please, find out revised manuscript.

Line 57- remove fibers and it doesn’t add anything

Response: Thanks for comment. We removed it (Line 58).

Line 61- I would argue that it is a very dynamic system, which changes constantly. So do you mean it is difficult to completely change? Or destroy? As in the main, it will alter in response to diet, stress, hormones, pregnancy etc

Response: Sorry for confusing. We revised this sentence to make clear (Line 61-62).

Line 66- maybe say growth stunting rather than disorders- but up to you

Response: We have replaced this word as followed by your suggestion. (Line 67)

Line 66- maybe including bvdv etc? Others include Cryptosporidium

Response: Thanks for suggestion. We revised this sentence (Line 67-69).

Line 69- link the two sentences

Response: We linked the two sentences according to your suggestion (Line 69-71).

Line 81-83- is this also diet associated? And stress?

Line 83- diarrhoeic calves?

Response: Thanks for comment. In introduction part, we revised this content as followed by your comment below.

Line 78-89- this is almost a summary and isn’t needed in the introduction. It needs an aims in here instead

Response: We have added aims of this study instead of summary in revised manuscript. (Line 79-82)

Line 98- was this colostrum from the same mother? Or commercial? If the latter please state manufacturer

Line 101- reword as doesn’t make sense

Response: Sorry for insufficient information. All neonatal calves fed same colostrum from different mother. We added additional information for colostrum feeding to revised manuscript. (Line 91-93)

Line 102- was this milk pasteurised or from the cows mother?

Line 103- reword as doesn’t make sense

Response: Thanks for comment. Calves fed non-pasteiurised whole milk from 3 days old. All calves same whole milk. We added additional information for milk feeding to revised manuscript. (Line 98)

Line 106- the grass on day 7 is likely to impact the microbiota, and could induce diarrhoea

Response: In our feeding protocol, mixed grass is provided to calves around 7 days old. Calves freely access to grass diet. But, grass intakes for the first and second weeks were very low. We agree that grass feeding is able to affect gut microbiota. But, all experimental calves have exposed to grass diet in a same manner.

Line 117-118- please reword as sounds repetitive

Response: We revised this sentence (Line 106-108)

Line 120- scoring systems

Response: We fixed error. (Line 109)

Line 124-127- this will also have affected the gut microbiota. You say that these were excluded, how many did that include? Based on the data below, 4 have diarrhoea, so they were removed, leaving 1 animal left?

Response: Sorry for confusing. We have excluded calves with fecal scores over 3 or received any treatments. Remained five experimental calves were healthy or show only mild diarrhea during experimental period.

Line 133- where in the sample you took this from is important as that will also affect microbiota

Response: Thanks for comments. Fecal sample were obtained from each calf using a swab sampling product. A sterile cotton swab was inserted mildly approximately 5 cm in the rectum and the samples were placed immediately in the enclosed tube containing preservatives. We added information in the revised manuscript. (Line 123-125)

Line 136- sampling every 7 days may miss the diarrhoea completely? So you see an alteration and miss it due to sampling technique?

Response: Thanks for valuable comment. Although, we collected fecal sample every 7 days, we had monitored fecal conditions for neonatal calves on a daily basis. We assigned particular sample as a diarrheal week sample, when calf showed diarrheal symptom in that week. Then, we attempted to compare fecal microbiome among pre-diarrhea, diarrhea, and post-diarrhea week.

Line 146-147- how was this further sequenced and analysed? More details needed

Response: Thanks for comments. Extracted DNA is normalized and pooled using the PicoGreen, and the size of libraries are verified using the TapeStation DNA screentape D1000 (Agilent). And then sequenced using the MiSeq™ platform. In M&M, further sequenced and analyzed information are shown in 2.5. A more detailed process is shown in the manuscript. (Line 138-167)

Why was 16S rRNA metagenomics chosen over shotgun? Diarrhoea may be of parasitic, protozoan or viral aetiology?

Response: We agree that shotgun sequencing is better option to detect other microbial signals beyond bacteria. However, we have focused to understand dynamic changes in fecal bacterial community and their functions by age and diarrhea in this study. We added limitation of our study in this aspect in revised manuscript (Line 528-532).

Line 184- this is the first mention of the 45 samples, maybe want to say where these came from.

Line 183-185- this may be useful in table with more info on the samples, numbers of sequences etc

Response: As followed by your suggestion, we added sample and sequence information in table 1 (Line 181).

Section 3.1 some figures and tables would make this much clearer to read

Response: Thanks for kind suggestion. We added Figure 1: Taxonomic composition of bacterial phyla and family using 16S rRNA gene sequencing to make easier for reader in revised manuscript.

Figure 3, this is a bit small to take much detail from, but that is probably a formatting issue

Response: Thanks for comment. We improved figures and table in revised manuscript.

Figure 1 looks great

Response: Thanks.

Line 214- is this OTU?

Response: Sorry for error. We fixed error (Line 217).

I think that most of the bacterial names need to be in italics

Response: Thanks for comment. We used italics at the species level and family level according to scientific nomenclature.

Line 256- reword

Response: We revised this sentence in manuscript (Line 260-262)

Line 263- please reword this as it sounds like the bacterium causes the disease rather than the family

Response: We modified this sentence (Line 266-268).

Line 267- I struggle with this as you only did 16S rRNA gene metagenomics, so how can you comment on functional genes?

Response: We have analyzed 16S rRNA metagenome data set by PICRUST to determine functional gene abundance. PICRUSt metagenome prediction algorithm is below. And also, we revised “functional genes” to “functional gene family” to avoid confusing. 1) When applying PICRUSt to a 16S rRNA gene library, PICRUSt matches reference operational taxonomic units against the tables, and retrieves a predicted 16S rRNA copy number and gene copy number for each gene family. 2) The abundance of each OTU is divided by its predicted copy number, and then multiplied by the copy number of the gene family. This gives a prediction for the contribution of each OTU to the overall gene content of the sample (the metagenome). 3) Individual contributions are summed together to produce an estimate of the genes present in the metagenome.

Line 301- would be nice to have more details on the gastroenteritis and when it occurs and in which animals, and if treated.

Response: Thanks for comment. We have added more detailed cause of diarrhea in calves in discussion section (Line 534-537).

Reviewer 2 Report

Thank you for your submission of this manuscript. It is strongly advised that the authors spend considerable time revising the current version of the manuscript. The authors need to consider tempering much of their tone about the findings of this study—this was an extremely small-scale study, with 4 of 5 calves developing diarrhea, so the normal microbiota is difficult to assess. Additionally, this study and the conclusions should be focused on a discussion of the fecal microbiota—the entire GI microbiota is not represented, and generalizations should not be made. The authors need to clearly define the objectives and aims of this study and then structure the Results and Discussions to mirror these objectives. The authors should also be careful about the use of the terms “microbiome” vs “microbiota” and should be specific throughout the manuscript (especially within the Introduction). Finally, the edits of this current version should be focused on increasing the readability and clinical impact of these findings. The portion of the Discussion focused on neonatal diarrhea is the most effective for the reader, but there are numerous other areas that should be improved.

Simple Summary

Line 19: Given that the authors only provide a single broad reference to “diseases” it may be more appropriate to expand or replace “such as” with “including”.

Lines 19-20: Please reword this sentence as it is confusing. Do the authors mean before in age or before in anatomic position?

Line 22: The word “reveal” is highly definitive; the authors should consider a different word given the breath of information still yet to understand regarding the microbiome.

Lines 26-29: Consider breaking this into two sentences—one in regards to the study and a second sentence about the larger impact of this work.

Abstract

Line 32: Again, the authors have made mention of “various factors” (ie a plural statement) but only provided a single example (ie diarrhea). Please reword.

Line 37: The authors should be careful about the use of the term “microbiome” since it would seem that “microbiota” is a more appropriate term (unless gene sequencing was performed comprehensively and not just an assessment of the observed microorganisms in the feces). Please edit these references carefully throughout the entire manuscript.

Line 45: Is there a “normal microbiome” for calves? If so, this should be discussed, otherwise consider revising since this evaluation is for similar calves in a similar geographical location at a similar time of year (ie multiple considerations here)

Line 50: Do the authors actually show there are differences in nutrient digestion? This should be revised to address that these are extrapolations of the findings.

Introduction

Lines 60-61: The authors have stated that the fecal microbiota changed rapidly with diarrhea, so this sentence (despite the references) is confusing. Additionally, the authors should address that fecal microbiota does not necessarily represent the entirety of the GI tract. This may be better addressed in the Discussion.

Line 69: Please consider adding an additional stress (not just diet change, but also environment, transport, etc), or revising plurality in this sentence.

Line 71 (and 74): These references (66-70 and 71-73) are out of order with the text. This is observed elsewhere in the manuscript, so please ensure that references are in order of presentation within the text. Additionally, it is important to recognize that these references are not just for ruminants but for monogastric animals as well. Please distinguish/expand.

Line 73: Please provide a reference for increased susceptibility to disease.

Lines 75-77: There are additional references about diarrhea and microbiota in animal species that the authors should reference here (and revise this sentence).

Line 81-86: This sentence makes a conclusion. The Introduction should state the study aims/objectives and hypotheses; with a final statement about the impact of such work. Please revise.

Materials and Methods

Lines 96: The number of calves (and related information) used should be included in the Results, not the M&M.

Line 100: Is this commercial colostrum? Whole milk? Please provide the source or manufacturer. Was any analysis performed to confirm appropriate passive transfer of IgG?

Line 105: Please provide the source information for the fed calf starter.

Table 1: This information may be appropriate as a supplementary item.

Lines 117-120: The term “we” is not typically used, please consult the guidelines by Animals

Line 133: Were fecal samples collected per rectum? Please specify. Additionally, where were samples taken from if a mass of feces were sampled—the surface or center? There has been some work (see Southwood as an author) about the location of the fecal sampling changing the microbial profile.

Line 162: Is may be helpful to provide brief descriptions of what is evaluated across these indices. Additionally, it may be valuable to provide a reference for these analytical techniques.

Results

Lines 202-204: Please provide a brief conclusion about the data displayed in Figure 1.

Line 205: What is meant by “other samples”?

Line 206: When the authors say “distinct” do they mean significant? If so, please provide a P value or explain what is meant but this term.

Line 208: “Gradually shifted” toward what?

Line 214: Should “OUT” be “OTU”? Please correct.

Lines 223-226: Please include a summary statement about the presented data of Figure 2 within the legend. Same comment for all additional Figure legends, please.

Line 267: It is unlikely that the fecal microbiota reflects the gut microbial composition. Please focus on speaking about the fecal microbial population only in the Results and address these potential differences in the Discussion.

Lines 275-279: Are asterisks visible for these comparisons? They are referenced in the legend by do not seem to be visible on Figure 4.

Lines 280-299: Please provide P values for significant comparisons.

Line 288: Grammatical error—edit “was” to “were”

Lines 315-318: The authors should state that these trends approached significance, given the provided P values.

Lines 322-324: Do these percentages account for cross-over genes? It would be helpful for the authors to remind the readers of the meanings of “level 1 and 2” to increase the impact and interpretation of these findings.

Line 328: Why are there underscores between these amino acids?

Discussion

Overall, the Discussion is difficult to follow, and should be reorganized to address each study aim/objective as presented in the original introduction and the Results. The paragraphs in the Discussion are lengthy and could be broken up to be more focused overall.

Line 356: Please provide a reference for the consideration of sterility of the calf stomach.

Lines 358-360: Please reference the bacterial community stages, as described.

Line 362: How does the colostrum administered in this study potentially affect the microbial communities based on this reference (8)? Was there any testing of the colostrum in this study? Additionally, please emphasize that distinct fecal microbial profiles were observed.

Line 380: The authors should emphasize that sika deer are relevant to mention as they have a similar GI tract structure to cattle. Additionally, was the referenced study looking at the rumen? This is likely not comparable directly to the fecal microbial profile discuss here for calves.

Line 397: Please define “early development”

Lines 402-404: This statement cannot be supported by this study given that the analysis was only evaluating the fecal microbial population and not the GI as a whole.

Line 410: Only a single reference is provided, although “metagenomic studies” (ie plural) are stated.

Lines 405-446: The authors present a number of different studies, but the presentation and comparison to the current study should be reorganized, as it is difficult for the reader to follow and to confirm the impact.

*The discussion of neonatal diarrhea and changes in the microbiota beginning on Line 448 has a high level of clinical relevance and impact for this paper and is a valuable discussion overall. The discussion of the analysis performed is also presented in a way that increases the accessibility for the reader.

Line 449: This is not always true—there may be a single etiologic agent in some cases.

Lines 453 and 457: Please provide citation(s) for these statements.

Line 459: Again, this sentence states “many studies” but only cites a single study—please revise.

Lines 483-484: Again, this study did not evaluate the intestinal microbiota, but fecal microbiota only.

Lines 524-525: Are the authors suggesting fecal transplantation? There are references about this technique that may be worth discussing.

Line 527-528: The authors should recognize that the findings from this study do not demonstrate whether the changes are a result of microbial shift, or microbial shift occurs due to diarrhea. Please discuss.

*The Discussion does not present any discussion of the limitations of this study, which absolutely are essential for inclusion.

Conclusions

*The Conclusions should directly response to the aims/objectives of the study, reference what needs to be addressed moving forward, and the overall impact of this word. Please consider restructuring this section.

References

Many of the references are formatted differently. Please review and be consistent with journal requirements.

Author Response

Response to Reviewer-2

Comments and Suggestions for Authors

Thank you for your submission of this manuscript. It is strongly advised that the authors spend considerable time revising the current version of the manuscript. The authors need to consider tempering much of their tone about the findings of this study—this was an extremely small-scale study, with 4 of 5 calves developing diarrhea, so the normal microbiota is difficult to assess. Additionally, this study and the conclusions should be focused on a discussion of the fecal microbiota—the entire GI microbiota is not represented, and generalizations should not be made. The authors need to clearly define the objectives and aims of this study and then structure the Results and Discussions to mirror these objectives. The authors should also be careful about the use of the terms “microbiome” vs “microbiota” and should be specific throughout the manuscript (especially within the Introduction). Finally, the edits of this current version should be focused on increasing the readability and clinical impact of these findings. The portion of the Discussion focused on neonatal diarrhea is the most effective for the reader, but there are numerous other areas that should be improved.

GENERAL RESPONSE:

Thanks for valuable comment. We agree that numbers of animals used in this study is a quite small. However, we used a small sample of animals (n = 5) for metagenome analysis, we observed highly clustered fecal microbiota patterns in PCoA analysis. This indicates that the samples in each group had small variations thus, identified OUT sequences were significant. We overall revised the manuscript and changed in the use of the term and described the limitations in discussion.

Also, we have improved our manuscript as followed by your kind suggestions. In revision, we focused on 1) Feeding information 2) Fecal microbiome vs Gut microbiome 4) Fecal microbiome vs. Gut microbiome, 4). References. Please, find out revised manuscript.

Simple Summary

Line 19: Given that the authors only provide a single broad reference to “diseases” it may be more appropriate to expand or replace “such as” with “including”.

Response: Thank you for your advice. We revised this sentence in the revised manuscript. (Line 19)

Lines 19-20: Please reword this sentence as it is confusing. Do the authors mean before in age or before in anatomic position?

Response: We revised this sentence to make clearer (Line 19-20).

Line 22: The word “reveal” is highly definitive; the authors should consider a different word given the breath of information still yet to understand regarding the microbiome.

Response: We revised “reveal” to “understand” in the revised manuscript (Line 22)

Lines 26-29: Consider breaking this into two sentences—one in regards to the study and a second sentence about the larger impact of this work.

Response: Thanks for suggestion. We broke them to two sentences to convey message more clearly in the revised manuscript. (Line 27-29)

Abstract

Line 32: Again, the authors have made mention of “various factors” (ie a plural statement) but only provided a single example (ie diarrhea). Please reword.

Response: We added more factors in the revised manuscript (Line 32-33)

Line 37: The authors should be careful about the use of the term “microbiome” since it would seem that “microbiota” is a more appropriate term (unless gene sequencing was performed comprehensively and not just an assessment of the observed microorganisms in the feces). Please edit these references carefully throughout the entire manuscript.

Response: Thanks for valuable comment. We understand your point. We put efforts to use appropriate word(s) for either microbiota or microbiome through checking whole manuscript. Please, find out a revised manuscript.

Line 45: Is there a “normal microbiome” for calves? If so, this should be discussed, otherwise consider revising since this evaluation is for similar calves in a similar geographical location at a similar time of year (ie multiple considerations here)

Response: Thanks for comment. We revised this sentence to make clearer (Line 45-46).

Line 50: Do the authors actually show there are differences in nutrient digestion? This should be revised to address that these are extrapolations of the findings.

Response: Thanks for comment. We edited this sentence with more appropriate words (Line 50-51).

Introduction

Lines 60-61: The authors have stated that the fecal microbiota changed rapidly with diarrhea, so this sentence (despite the references) is confusing. Additionally, the authors should address that fecal microbiota does not necessarily represent the entirety of the GI tract. This may be better addressed in the Discussion.

Response: Thanks for comment. We have modified thins sentence. Additionally, we have added discussion for this issue: difference between fecal microbiome and gut microbiome in revised manuscript (Line 529-532).

Line 69: Please consider adding an additional stress (not just diet change, but also environment, transport, etc), or revising plurality in this sentence.

Response: Thanks for suggestion. We added more examples (Line 69-71).

Line 71 (and 74): These references (66-70 and 71-73) are out of order with the text. This is observed elsewhere in the manuscript, so please ensure that references are in order of presentation within the text. Additionally, it is important to recognize that these references are not just for ruminants but for monogastric animals as well. Please distinguish/expand.

Response: Sorry for error. We worked for reference formatting issue during revision.

Line 73: Please provide a reference for increased susceptibility to disease.

Lines 75-77: There are additional references about diarrhea and microbiota in animal species that the authors should reference here (and revise this sentence).

Response: Thanks for comment. We have added appropriate reference (Line 74) (Line 78).

Line 81-86: This sentence makes a conclusion. The Introduction should state the study aims/objectives and hypotheses; with a final statement about the impact of such work. Please revise.

Response: We revised last part of introduction as followed by your suggestion (Line 79-82).

Materials and Methods

Lines 96: The number of calves (and related information) used should be included in the Results, not the M&M.

Response: Thanks for comment. We added information for numbers of calves in results part as well.

Line 100: Is this commercial colostrum? Whole milk? Please provide the source or manufacturer. Was any analysis performed to confirm appropriate passive transfer of IgG?

Response: Thanks for comment. Calves fed same whole milk. We added additional information to revised manuscript. (Line 97-99). In order to make sure for proper passive transfer of IgG through colostrum, we measured IgG of colostrum (66.46±27.53 g/L).

Line 105: Please provide the source information for the fed calf starter.

Response: We added information for calf starter (Line 100-101).

Table 1: This information may be appropriate as a supplementary item.

Response: We moved “Table 1” to supplementary information.

Lines 117-120: The term “we” is not typically used, please consult the guidelines by Animals

Response: Thank you for valuable comment. We revised sentence as followed by journal’s guideline (Line 106).

Line 133: Were fecal samples collected per rectum? Please specify. Additionally, where were samples taken from if a mass of feces were sampled—the surface or center? There has been some work (see Southwood as an author) about the location of the fecal sampling changing the microbial profile.

Response: Thanks for valuable comment. We agree that fecal sample collection method is critical. We added detailed information for fecal sample collection in revised manuscript (Line 123-125).

Line 162: Is may be helpful to provide brief descriptions of what is evaluated across these indices. Additionally, it may be valuable to provide a reference for these analytical techniques.

Response: Thank you for your advice and we added a reference according to your advice. (Line 151-152)

Results

Lines 202-204: Please provide a brief conclusion about the data displayed in Figure 1.

Response: Thanks for comment. We added more conclusive heading of legend for figure 1.

Line 205: What is meant by “other samples”?

Response: Apologize for error. We edited this sentence in the revised manuscript. (Line 208)

Line 206: When the authors say “distinct” do they mean significant? If so, please provide a P value or explain what is meant but this term.

Response: Sorry to make you confused. This sentence refers to the feces microbiota group of PCoA shown in Figure 1. 'Distinct' means that unlike the density of each group of samples and the group in Week 1 was separated from the other groups.

Line 208: “Gradually shifted” toward what?

Response: We reworded for this sentence (Line 211-212)

Line 214: Should “OUT” be “OTU”? Please correct.

Response: Apologize. We fixed error (Line 217).

Lines 223-226: Please include a summary statement about the presented data of Figure 2 within the legend. Same comment for all additional Figure legends, please.

Response: Thanks for comment. We added more conclusive heading of legend for Figure 2 (not Figure 3).

Line 267: It is unlikely that the fecal microbiota reflects the gut microbial composition. Please focus on speaking about the fecal microbial population only in the Results and address these potential differences in the Discussion.

Response: Thanks for comment. We tried to use term(s) more carefully and added discussion for this issue in revised manuscript (Line 529-532).

Lines 275-279: Are asterisks visible for these comparisons? They are referenced in the legend by do not seem to be visible on Figure 4.

Response: Sorry for confusing. We found that there was an error in the description of Figure 4 and corrected it. We revised legend of Figure 4 and Figure 6 (Line 279-286, 349-356)

Lines 280-299: Please provide P values for significant comparisons.

Response: We were sorry to make you confused. Figure 4 shown all the pathways, which significantly different (P<0.05).

Line 288: Grammatical error—edit “was” to “were”

Response: We apologize for error. We revised them in the revised manuscript. (Line 295)

Lines 315-318: The authors should state that these trends approached significance, given the provided P values.

Response: Thanks for comment. We revised this sentence in revised manuscript. (Line 321-323)

Lines 322-324: Do these percentages account for cross-over genes? It would be helpful for the authors to remind the readers of the meanings of “level 1 and 2” to increase the impact and interpretation of these findings.

Response: We added more information for Level 1 for reader. (Line 330-332)

Line 328: Why are there underscores between these amino acids?

Response: We revised them in the revised manuscript. (Line 338)

Discussion

Overall, the Discussion is difficult to follow, and should be reorganized to address each study aim/objective as presented in the original introduction and the Results. The paragraphs in the Discussion are lengthy and could be broken up to be more focused overall.

Response: Thanks for comment. We agree that discussion in original manuscript had many issues. So, we tried put efforts to improve discussion part during revision.

Lines 358-360: Please reference the bacterial community stages, as described.

Response: We revised this sentence in manuscript (Line 370-372)

Line 362: How does the colostrum administered in this study potentially affect the microbial communities based on this reference (8)? Was there any testing of the colostrum in this study? Additionally, please emphasize that distinct fecal microbial profiles were observed.

Response: We added more detailed information for colostrum feeding used in this study. We also tried to emphasize distinct fecal microbial profiles, probably caused by colostrum.

Line 380: The authors should emphasize that sika deer are relevant to mention as they have a similar GI tract structure to cattle. Additionally, was the referenced study looking at the rumen? This is likely not comparable directly to the fecal microbial profile discuss here for calves.

Response: Thanks for comment. We added appropriate reference in revised manuscript (Line 391 – 394).

Line 397: Please define “early development”

Response: We added detailed information for that reference (Line 410-411)

Lines 402-404: This statement cannot be supported by this study given that the analysis was only evaluating the fecal microbial population and not the GI as a whole.

Lines 483-484: Again, this study did not evaluate the intestinal microbiota, but fecal microbiota only.

Response: Thanks for comment. We understand your point thus, we revised discussion with separation between fecal microbiome and gut microbiome. Additionally, we have added issue on difference between fecal microbiome and gut microbiome in discussion part. (Line 528-532). Please, find out revised manuscript.

Lines 405-446: The authors present a number of different studies, but the presentation and comparison to the current study should be reorganized, as it is difficult for the reader to follow and to confirm the impact.

Response: Thanks for valuable comment. We revised this section to make better discussion with providing connection between references and our study. Please, find out revised manuscript (Line 419-430).

*The discussion of neonatal diarrhea and changes in the microbiota beginning on Line 448 has a high level of clinical relevance and impact for this paper and is a valuable discussion overall. The discussion of the analysis performed is also presented in a way that increases the accessibility for the reader.

Response: Thanks for valuable comment. We had put more efforts to strengthen discussion regarding this issue.

Line 449: This is not always true—there may be a single etiologic agent in some cases.

Response: We meant various factors are able to induce diarrheal disease in neonatal calves.

Line 410: Only a single reference is provided, although “metagenomic studies” (ie plural) are stated.

Response: Thanks for comment. We deleted this content as followed by your comment below

Line 356: Please provide a reference for the consideration of sterility of the calf stomach.

Lines 453 and 457: Please provide citation(s) for these statements.

Line 459: Again, this sentence states “many studies” but only cites a single study—please revise.

Response: We have added appropriate references (s) for these sentences (Line 368) (Line 468) (Line 471).

Lines 524-525: Are the authors suggesting fecal transplantation? There are references about this technique that may be worth discussing.

Response: Thanks for suggestion. We agree that discussing FMT is too far from our study. We deleted this content and added more valuable discussions such as limitation of our study (Line 530-543).

527-528: The authors should recognize that the findings from this study do not demonstrate whether the changes are a result of microbial shift, or microbial shift occurs due to diarrhea. Please discuss.

Response: Thanks for valuable comment. We already recognized this issue and mentioned about it in original manuscript. However, we added more discussion for this issue in revised manuscript (Line 461-465).

*The Discussion does not present any discussion of the limitations of this study, which absolutely are essential for inclusion.

Response: Thanks for comment. We agree that our study have several limitations. We added contents for limitation of current study to make more valuable discussion (Line 528-542).

Conclusions

*The Conclusions should directly response to the aims/objectives of the study, reference what needs to be addressed moving forward, and the overall impact of this word. Please consider restructuring this section.

Response: Thanks for comment. We revised whole conclusion part to covey message from our finding more clearly to reader. Please find it in revised manuscript (Line 551-563).

References

Many of the references are formatted differently. Please review and be consistent with journal requirements.

Response: We formatted all references again.

Round 2

Reviewer 1 Report

The manuscript is much improved, however some changes to the grammar need to be made prior to acceptance. I have detailed these below, with what the manuscript should read as to aid the authors. I thank you for talking on board my comments previously, and wish you all the best for the future.

Line 61- The established gut microbiome has a significant impact (reword)

Line 69-70- Neonatal calves also suffer from diarrhea caused by non-infectious stresses, (reword)

Line 79- The aim of the present study is to understand changes of diversity and function in fecal (reword)

Line 91- We obtained colostrum from mother(s) and mixed. Mothers of the calves? Mothers in general? What ratio etc. Needs more information

Line 92- All experimental neonatal calves were fed same colostrum. (reword)

Line 98- We obtained whole milk from mother(s) and mixed. – as above, same mothers of calves? Mixed in what ratio? Bit more information needed

Line 98- Neonatal calves were fed the same whole milk. (reword)

Line 122-125- this doesn’t make sense and needs rewording. It sounds like you have taken the rectum?

Line 153-154- Briefly, Chao1 is used to estimate diversity from the abundance number of species in a community and the Shannon index is used to quantify specific biodiversity. (reword)

Line 178- change range to ranged

Table 1- some information on what the sample name means may be good

Line 261- replace differ with different

Line 370- The primary phase starts on day 2, and is mainly composed of Proteobacteria (reword)
Line 392-393- richness with age in fecal microbiome of dairy calves, regardless of weaning (reword)

Line 397-  Although we do point out the major single reason for the drastic shift ….(reword)

Line 420 - However, there are some inconsistent reports for the most dominant gut microbiome at the phylum level- this is unclear. I think you mean the most dominant bacteria in the gut microbiome?

427-428- There were no considerable differences in abundance found at the phyla level during the pre-weaning stage. (reword)

Line 429- Unlike our study, Dias et al. [45] showed Bacteroidetes increase in the colon and cecum as the calves aged before weaning (reword)

Line 468-469- Therefore, the unstable and low diversity of microbiome is likely to cause diarrhea and needs to be careful.- reword as doesn’t make sense. Need to be careful with what?

Line 497- Thus, having a normal range of Bacteroides may be reliable indicator for a healthy gut

Line 529- replace study with studies

Line 533- we could not point out a major reason for the dramatic changes of fecal microbiome (reword)

Line 534- As a variety of factors affect fecal microbiome (reword)

Line 536- but a dietary intervention experiment is needed to be performed in future studies (reword).

Line 537- Also, shotgun metagenomic sequencing may help to find causative agents such as viruses for the calf diarrhea (reword)

Line 540- . It seems that the gut ecosystem….

Line 554- which is characterized by the high (reword)

Author Response

Response to Reviewer-1

Line 61- The established gut microbiome has a significant impact (reword)

Line 69-70- Neonatal calves also suffer from diarrhea caused by non-infectious stresses, (reword)

Line 79- The aim of the present study is to understand changes of diversity and function in fecal (reword)

Line 92- All experimental neonatal calves were fed same colostrum. (reword)

Line 98- Neonatal calves were fed the same whole milk. (reword)

Line 153-154- Briefly, Chao1 is used to estimate diversity from the abundance number of species in a community and the Shannon index is used to quantify specific biodiversity. (reword)

Line 178- change range to ranged

Line 261- replace differ with different

Line 370- The primary phase starts on day 2, and is mainly composed of Proteobacteria (reword)

Line 392-393- richness with age in fecal microbiome of dairy calves, regardless of weaning (reword)

Line 397- Although we do point out the major single reason for the drastic shift ….(reword)

427-428- There were no considerable differences in abundance found at the phyla level during the pre-weaning stage. (reword)

Line 429- Unlike our study, Dias et al. [45] showed Bacteroidetes increase in the colon and cecum as the calves aged before weaning (reword)

Line 468-469- Therefore, the unstable and low diversity of microbiome is likely to cause diarrhea and needs to be careful.- reword as doesn’t make sense. Need to be careful with what?

Line 497- Thus, having a normal range of Bacteroides may be reliable indicator for a healthy gut

Line 529- replace study with studies

Line 533- we could not point out a major reason for the dramatic changes of fecal microbiome (reword)

Line 534- As a variety of factors affect fecal microbiome (reword)

Line 536- but a dietary intervention experiment is needed to be performed in future studies (reword).

Line 537- Also, shotgun metagenomic sequencing may help to find causative agents such as viruses for the calf diarrhea (reword)

Line 540- . It seems that the gut ecosystem….

Line 554- which is characterized by the high (reword)

Response: Thanks for kind corrections. We edited these sentences as followed by your suggestion. Please, find indicated area in the revised manuscript (Line 61, 69-71, 78, 92-93, 99, 153-154, 180, 264, 383-384, 406-409, 439-441, 481-482, 508-509, 546, 552-553, 553-554, 556-557, 563-566, 559, 577).

Line 91- We obtained colostrum from mother(s) and mixed. Mothers of the calves? Mothers in general? What ratio etc. Needs more information

Line 98- We obtained whole milk from mother(s) and mixed. – as above, same mothers of calves? Mixed in what ratio? Bit more information needed

Response: Thanks for comment. We obtained mixed the colostrum and milk from the general mother(s) cow. All experimental calves fed same amount and quality of colostrum and milk. We added a little more detailed information for colostrum and milk used in this study during revision. (Line 91-92), (Line 98-99)

Line 122-125- this doesn’t make sense and needs rewording. It sounds like you have taken the rectum?

Response: Sorry for confusion. We revised this sentence. “Calf fecal samples were collected using a sterile swab kit (Fecal swab collection and preservation system, Norgen Biotek, Ontario, Canada) once a week for 8 weeks. Fecal samples were collected from approximately 5 cm in the rectum area and the samples were placed immediately in the enclosed tube containing preservative.” (Line 122-125).

Table 1- some information on what the sample name means may be good

Response: We corrected sample name and provide information for sample name; Animals number.week of old (Line 184)

Line 420 - However, there are some inconsistent reports for the most dominant gut microbiome at the phylum level- this is unclear. I think you mean the most dominant bacteria in the gut microbiome?

Response: We made this sentence clearer: “Bacteriodetes, Firmicutes, or Proteobacteria are the major phyla found in cows from several studies regardless of age. However, there are some inconsistent reports for the most dominant gut bacteria at the phylum level.” (Line 431-433)

Reviewer 2 Report

Comments to the authors:

Thank you for your thoughtful revisions of this manuscript; these edits have improved the clarity. There are still areas where the authors make strong statements about the changes observed, and fail to recognize the small number of animals in the study. Please also consider providing an improved structure to the Discussion, as it will have a greater impact to the reader. Thank you.

Simple Summary

The edits have improved the clarity of this summary.

Abstract

Lines 45-46: Would be helpful to mention how quickly this return to pre-diarrheal stage occurred.

Line 47: It is assumed that this decrease is significant (since it is stated), and “significantly” can be removed from the sentence. The authors may choose to provide a P value after “decreased”.

Introduction

Line 58: The term “the main feed” is a bit vague, consider being more specific.

Lines 69-70: Please check grammar of sentence beginning with “Neonatal calves are also…”

Lines 77-78: Is this sentence correct? Are the authors suggesting this is true across the literature or only in calves? Please specify, since more references are likely available if all species are being considered.

Materials & Methods

Lines 91-93: Was the IgG of these calves measured to ensure that passive transfer of immunoglobulins was achieved through administration of colostrum?

Lines 101-102: Why have the authors mentioned the feeding of orchard hay when the study is focused on the pre-weaning phase? This would likely cause a substantial shift in the microbiota.

Line 154: The word “used” does not need to be capitalized.

*Overall the description and organization of the Materials & Methods in this revised manuscript version is substantially improved.

Lines 171-172: Was normality of these data evaluated? With such a small sample size, it seems surprising that the assumptions for normality were met. Please provide additional clarification on this assessment and how data was treated (i.e. independent samples at each time point vs repeated measures).

Results

*The authors should carefully edit the Results under the guidelines of the journal. This Reviewer feels it is redundant to state “significantly increased/decreased”in the Results, since it is expected that the authors would not state non-significant data. Because of this, it is likely more appropriate to state that there is a difference (i.e. increase or decreased) plus the P value.

Lines 200-202: Although the PCoA graph is used for understanding the microbial composition transition, this sentence should be revised in the manuscript text to make a statement—even a general one—about this transition, instead of simply referring to Figure 2.

Lines 211: What is meant by “microbial composition group”?

Line 218: The term “significant” is not needed in this sentence.

Line 245: Wasn’t the entire study focused in the pre-weaning period for these calves? Do the authors mean the early vs late pre-weaning period? Please comment in the Discussion about some of these bacterial shifts and how they may correspond to changes in the feeding plan.

Line 256: Please explain what is meant by the “main contributors”? Are the authors suggesting that these species caused the change? Or rather that they were most substantially affected? Please clarify.

Line 261: Please change “differ” to “different”. Additionally, the term “significantly” is not needed.

Line 265: Please clarify what is meant by “significant members”.

Line 267: Please clarify what is meant by “appeared significantly”.

Line 268: The authors state that the fecal microbiota “changed gradually” but discuss specific species appearing acutely. Do these species gradually increase after their appearance? Please ensure the potential sources for these changes (milk, environment, etc) in the Discussion.

Line 285: Please provide a reference for the Hutlab Galaxy

Lines 288-289: Please remove the term “significantly”, and state the P value.

Line 292-306: Anytime a comparison is being made (ie “a decrease at week 3”) a P value should be stated. There are multiple sections further in the Results section where this comment also applies—please double-check all sentences.

Line 308: The sentence with reference 40 should not be in the Results—please move to the Discussion.

Line 322: Please revise the grammar of the phrase “did not have significance”.

Line 353: In the legend for Figure 6, it says a Kruskal-Wallis test was used to determine significance, however there are no references in the figure or text to the P values. Please revise.

Discussion

Lines 372-374: Were any calves in this study born via C-section? Please state in the Methods if all were delivered vaginally.

Line 385: Do the authors mean breastfeeding? Or simply colostrum administration?

Line 397: Please reword this sentence—it is unclear what the major single reason is? Or are the authors saying they do not identify a major, single reason for this shift?

Lines 408-411: Why did the authors select 8 weeks?

Lines 458-476: Do the authors feels that this study accurately represents the normal microbiota of the neonatal GI? Given that such a large proportion of calves developed diarrhea, can the authors discuss why this might have occurred? Could it be that the microbial profile observed here is actually the reason these calves developed diarrhea? The authors do state they don’t know in lines 538-540, but they authors are quite direct in stating the importance of these changes observed, so it would be helpful for the reader for the authors to either revise the tone of their Discussion or provide a greater amount of interpretation to the reader.

Conclusions

*The authors state throughout the Discussion and Conclusion about the “drastic changes”, but this isn’t really supported by the presented data. The authors have not tried to relate these changes to changes in feed and should do so for the reader.

Author Response

Response to Reviewer-2

Thank you for your thoughtful revisions of this manuscript; these edits have improved the clarity. There are still areas where the authors make strong statements about the changes observed, and fail to recognize the small number of animals in the study. Please also consider providing an improved structure to the Discussion, as it will have a greater impact to the reader. Thank you.

Response: Thank you for comments. We agreed with your comment. We also recognized limitation of our study but, believe that our study still has meaningful information in this field. In second round of revision, we put more efforts to avoid overstatement for our findings from this study and re-organized discussion part to convey message more clearly. Please, find changes in the revised manuscript.

Simple Summary

The edits have improved the clarity of this summary.

Response: Thanks

Abstract

Lines 45-46: Would be helpful to mention how quickly this return to pre-diarrheal stage occurred.

Response: Thanks for comment. We added information duration to present how quickly returned to base level (1 week). (Line 45-46)

Line 47: It is assumed that this decrease is significant (since it is stated), and “significantly” can be removed from the sentence. The authors may choose to provide a P value after “decreased”.

Response: We have deleted ‘significant’ in this sentence and provided a p-value for this comparison. (Line 46-47)

Introduction

Line 58: The term “the main feed” is a bit vague, consider being more specific.

Response: We have changed to “macronutrients such as carbohydrates, lipids, proteins”. (Line 58)

Lines 69-70: Please check grammar of sentence beginning with “Neonatal calves are also…”

Response: Thanks for comment. We re-wrote this sentence. “In other hands, non-infectious stresses such as changes in diet and breeding facility induce diarrheal diseases in neonatal calves.” (Line 69-71)

Lines 77-78: Is this sentence correct? Are the authors suggesting this is true across the literature or only in calves? Please specify, since more references are likely available if all species are being considered.

Response: Thanks for comment. We have revised this sentence: “However, we do not fully understand for biological connection between diarrhea and gut microbiome in calves.” (Line 76-77)

Materials & Methods

Lines 91-93: Was the IgG of these calves measured to ensure that passive transfer of immunoglobulins was achieved through administration of colostrum?

Response: Thanks for comment. Unfortunately, we could not have opportunity to examine blood IgG levels in neonatal calves with colostrum feeding. However, we have confirmed that colostrum used in this study had normal range of total IgG through ELISA assay. Thus, we think that neonatal calves were likely to have normal passive immunity in this study.

Lines 101-102: Why have the authors mentioned the feeding of orchard hay when the study is focused on the pre-weaning phase? This would likely cause a substantial shift in the microbiota.

Response: We provide feed to experimental calves according to calf feeding guideline by our institution.

Line 154: The word “used” does not need to be capitalized.

Response: We have fixed error. (Line 153)

*Overall the description and organization of the Materials & Methods in this revised manuscript version is substantially improved.

Lines 171-172: Was normality of these data evaluated? With such a small sample size, it seems surprising that the assumptions for normality were met. Please provide additional clarification on this assessment and how data was treated (i.e. independent samples at each time point vs repeated measures).

Response: Thanks for valuable comment. We performed statistical analysis again and there was some minor changes. Data for OTUs, Chao1, Shannon and Inverse Simpson were analyzed by one-way ANOVA with Tukey’s multiple comparison test after normality test. However, data for bacterial composition in figure 7 were analyzed by Kruskall-Wallis and Dunn’s multiple comparison test as they did not meet the normality test. We provided statistical analysis information in revised manuscript. (Line 169-174)

Results

*The authors should carefully edit the Results under the guidelines of the journal. This Reviewer feels it is redundant to state “significantly increased/decreased”in the Results, since it is expected that the authors would not state non-significant data. Because of this, it is likely more appropriate to state that there is a difference (i.e. increase or decreased) plus the P value.

Lines 200-202: Although the PCoA graph is used for understanding the microbial composition transition, this sentence should be revised in the manuscript text to make a statement—even a general one—about this transition, instead of simply referring to Figure 2.

Response: Thanks for comment. We revised this sentence: “The microbial composition transition from weeks 0 to 8 was represented by three-dimensional principal coordinates analysis (PCoA) in Figure 2”. (Line 203-205)

Lines 211: What is meant by “microbial composition group”?

Response: We corrected this sentence: “After 1 week, the composition of gut bacteria gradually changed until 6 weeks of age”. (Line 214-215)

Line 218: The term “significant” is not needed in this sentence.

Response: We deleted this in the sentence. (Line 220-221)

Line 245: Wasn’t the entire study focused in the pre-weaning period for these calves? Do the authors mean the early vs late pre-weaning period? Please comment in the Discussion about some of these bacterial shifts and how they may correspond to changes in the feeding plan.

Response: Thanks for valuable comment. We put more efforts on providing information for connection between bacteria shift and feeding plan used in this study. Please, find changes in discussion part.

Line 256: Please explain what is meant by the “main contributors”? Are the authors suggesting that these species caused the change? Or rather that they were most substantially affected? Please clarify.

Response: Thanks for comment. We revised this sentence more clearly: “The listed bacteria in Figure 4C were the main bacterial species to explain the changes in the corresponding families.” (Line 258-259)

Line 261: Please change “differ” to “different”. Additionally, the term “significantly” is not needed.

Response: We fixed error. (Line 264)

Line 265: Please clarify what is meant by “significant members”.

Response: We revised sentence: “In addition, Lactobacillus johnsonii, one of the important probiotic strains did not show a high relative abundance”. (Line 268-269)

Line 267: Please clarify what is meant by “appeared significantly”.

Response: We revised this sentence to present our observation more clearly.; “Escherichia fergusonii, a species of the Enterobacteriaceae family, appeared clearly at week 4 but they disappear from weeks 6.” (Line 269-270)

Line 268: The authors state that the fecal microbiota “changed gradually” but discuss specific species appearing acutely. Do these species gradually increase after their appearance? Please ensure the potential sources for these changes (milk, environment, etc) in the Discussion.

Response: We revised the sentence.; “These results showed that the composition of fecal bacteria of calves changed in certain periods.” (Line 270-271) We agree that dietary factor is important for gut microbiome changes. We tried to provide more information for connection between bacteria shift and feeding plan used in this study. However, there are limited information for fecal microbiome changes by dietary intervention.

Line 285: Please provide a reference for the Hutlab Galaxy

Line 353: In the legend for Figure 6, it says a Kruskal-Wallis test was used to determine significance, however there are no references in the figure or text to the P values. Please revise.

Response: We added reference for this information. (Line 288, 345)

Lines 288-289: Please remove the term “significantly” and state the P value.

Line 292-306: Anytime a comparison is being made (ie “a decrease at week 3”) a P value should be stated. There are multiple sections further in the Results section where this comment also applies—please double-check all sentences.

Response: We deleted ‘significant’ and added p-value for the entire manuscripts. (Line 297, 299, 302, 307)

Line 308: The sentence with reference 40 should not be in the Results—please move to the Discussion.

Response: We have deleted this sentence in result section.

Line 322: Please revise the grammar of the phrase “did not have significance”.

Response: We corrected this sentence: “The relative abundance of Bacteroidaceae recovered after diarrhea (p = 0.33).” (Line 330-331) 

Discussion

Lines 372-374: Were any calves in this study born via C-section? Please state in the Methods if all were delivered vaginally.

Response: Thanks for comment. We added more information for delivery method. (Line 88-89)

Line 385: Do the authors mean breastfeeding? Or simply colostrum administration?

Response: Thanks for comment. it means breastfeeding, not colostrum administration.

Line 397: Please reword this sentence—it is unclear what the major single reason is? Or are the authors saying they do not identify a major, single reason for this shift?

Response: We revised sentence to make clearer: “Although we could not point out single reason for the drastic shift in fecal microbiome during early week of life, we postulate that colostrum, vaginal birth, salivary microorganisms, and the microbiota of the dams contribute to making the difference at week 0.” (Line 406-409)

Lines 408-411: Why did the authors select 8 weeks?

Response: We thought this period was very important in determining microbiome changes in growth and gut, so we chose 0-8 weeks.

Lines 458-476: Do the authors feels that this study accurately represents the normal microbiota of the neonatal GI? Given that such a large proportion of calves developed diarrhea, can the authors discuss why this might have occurred? Could it be that the microbial profile observed here is actually the reason these calves developed diarrhea? The authors do state they don’t know in lines 538-540, but they authors are quite direct in stating the importance of these changes observed, so it would be helpful for the reader for the authors to either revise the tone of their Discussion or provide a greater amount of interpretation to the reader.

Response: Thanks for valuable comments. We agree that this issue provides important insights for readers in the field of animal health. We added more discussion, including potential limitation of study and microbial shift by diarrhea. Please, find changes in a revised manuscript.

Conclusions

*The authors state throughout the Discussion and Conclusion about the “drastic changes”, but this isn’t really supported by the presented data. The authors have not tried to relate these changes to changes in feed and should do so for the reader.

Response: Thanks for comments. We revised discussion and conclusion carefully and tried to discuss feed change as a potential cause of fecal microbiome.
